# Temporally Consistent Video Transformer for Long-Term Video Prediction

## Abstract

Generating long, temporally consistent video remains an open challenge in video generation. Primarily due to computational limitations, most prior methods limit themselves to training on a small subset of frames that are then extended to generate longer videos through a sliding window fashion. Although these techniques may produce sharp videos, they have difficulty retaining long-term temporal consistency due to their limited context length. In this work, we present **Te**mporally **Co**nsistent Video Transformer (TECO), a vector-quantized latent dynamics video prediction model that learns compressed representations to efficiently condition on long videos of hundreds of frames during both training and generation. We use a MaskGit prior for dynamics prediction which enables both sharper and faster generations compared to prior work. Our experiments show that TECO outperforms SOTA baselines in a variety of video prediction benchmarks ranging from simple mazes in DMLab, large 3D worlds in Minecraft, and complex real-world videos from Kinetics-600. In addition, to better understand the capabilities of video prediction models in modeling temporal consistency, we introduce several challenging video prediction tasks consisting of agents randomly traversing 3D scenes of varying difficulty. This presents a challenging benchmark for video prediction in partially observable environments where a model must understand what parts of the scenes to re-create versus invent depending on its past observations or generations. Generated videos are available on the website: https://sites.google.com/view/iclr23-teco

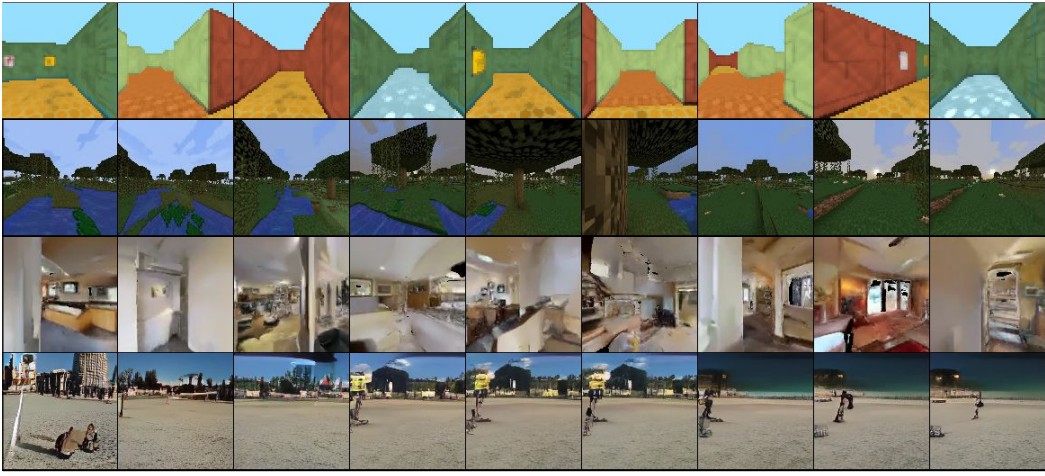

t = 36                                                                299

Figure 1: TECO generates sharp and consistent video predictions for hundreds of frames on challenging datasets. The figure shows evenly spaced frames of the 264 frame predictions, after being conditioned on 36 context frames. From top to bottom, the datasets are are DMLab, Minecraft, Habitat, and Kinetics-600.

## 1 INTRODUCTION

Recent work in video prediction has seen tremendous progress (Ho et al., 2022; Clark et al., 2019; Yan et al., 2021; Le Moing et al., 2021; Ge et al., 2022; Tian et al., 2021; Luc et al., 2020) in producing high-fidelity and diverse samples on complex video data. This can largely be attributed to a combination of increased computational resources and more compute efficient high-capacity neural architectures. However, much of this progress has focused on generating short videos, where models can perform well by basing their predictions on only a handful of previous frames.

Video prediction models with short context windows can generate long videos in a sliding window fashion. While the resulting videos can look impressive at first sight, they lack temporal consistency. We would like models to predict temporally consistent videos — where the same content is generated if a camera pans back to a previously observed location. On the other hand, the model should imagine a new part of the scene for locations that have not yet been observed, and future predictions should remain consistent to this newly imagined part of the scene.

Prior work has investigated techniques for modeling long-term dependencies, such as temporal hierarchies (Saxena et al., 2021) and strided sampling with frame-wise interpolation (Ge et al., 2022; Hong et al., 2022). Other methods train on sparse sets of frames selected out of long videos (Harvey et al., 2022; Skorokhodov et al., 2021; Clark et al., 2019; Saito & Saito, 2018; Yu et al., 2022), or model videos via compressed representations (Yan et al., 2021; Rakhimov et al., 2020; Le Moing et al., 2021; Seo et al., 2022; Gupta et al., 2022; Walker et al., 2021). Refer to Appendix M for more detailed discussion on related work.

Despite this progress, many methods still have difficulty scaling to datasets with many long-range dependencies. While Clockwork-VAE (Saxena et al., 2021) trains on long sequences, it is limited by training time (due to a recurrent architecture) and difficult to scale to more complex data. On the other hand, transformer-based methods over latent spaces (Yan et al., 2021) scale poorly to long videos due to quadratic complexity in attention, with long videos containing tens of thousands of tokens. Methods that train on subsets of tokens are limited by truncated backpropagation through time (Hutchins et al., 2022; Rae et al., 2019; Dai et al., 2019) or naive temporal operations (Hawthorne et al., 2022).

In this paper, we introduce **Te**mporally **Co**nsistent Video Transformer (TECO), a vector-quantized latent dynamics model that effectively models long-term dependencies in a compact representation space using efficient transformers. The key contributions are summarized as follows:

- We introduce TECO, an efficient and scalable video prediction model that learns a set of compressed VQ-latents to allow for efficient training and generation.

- We propose several long-length video prediction datasets centered around 3D scenes in DMLab (Beattie et al., 2016), Minecraft (Guss et al., 2019), and Habitat (Szot et al., 2021; Savva et al., 2019) to help better evaluate temporal consistency in video predictions.

- We show that TECO has strong performance on a variety of difficult video prediction tasks, and is able to leverage long-term temporal context to generate high quality videos with consistency.

- We provide several ablations providing intuition for why TECO is able to generate more temporally consistency predictions, and how these insights can extend to future work in long-term video prediction.

## 2 PRELIMINARIES

### 2.1 VQ-GAN

VQ-GAN (Esser et al., 2021; Van Den Oord et al., 2017) is an autoencoder that learns to compress data into a set of discrete latents, consisting of an encoder $E$, decoder $G$, codebook $C$, and discriminator $D$. Given an image $x \in \mathbb{R}^{H \times W \times 3}$, the encoder $E$ maps $x$ to its latent representation $h \in \mathbb{R}^{H' \times W' \times D}$, which is quantized by nearest neighbors lookup in a codebook of embeddings $C = \{e_i\}_{i=1}^{K}$ to produce $z \in \mathbb{R}^{H' \times W' \times D}$. The discretized latent $z$ is fed through decoder $G$ to

reconstruct $x$. A straight-through estimator (Bengio, 2013) is used to maintain gradient flow through the quantization step. The codebook optimizes the following loss:

$$\mathcal{L}_{\text{VQ}} = \| \operatorname{sg}(h) - e \|_2^2 + \beta \| h - \operatorname{sg}(e) \|_2^2 \tag{1}$$

where $\beta = 0.25$ is a hyperparameter, and $e$ is the corresponding nearest-neighbors embedding from codebook $C$. For reconstruction, VQ-GAN replaces the original $\ell_2$ loss with a perceptual loss (Zhang et al., 2012), $\mathcal{L}_{\text{LPIPS}}$. Finally, in order to encourage higher-fidelity samples, patch-level discriminator $D$ is trained to classify between real and reconstructed images, with.

$$\mathcal{L}_{\text{GAN}} = \log D(x) + \log(1 - D(\hat{x})) \tag{2}$$

Overall, VQ-GAN optimizes the following combination of losses:

$$\min_{E,G,C} \max_{D} \ \mathcal{L}_{\text{LPIPS}} + \mathcal{L}_{\text{VQ}} + \lambda \mathcal{L}_{\text{GAN}} \tag{3}$$

where $\lambda = \frac{\| \nabla_{G_L} \mathcal{L}_{\text{LPIPS}} \|_2}{\| \nabla_{G_L} \mathcal{L}_{\text{GAN}} \|_2 + \delta}$ is an adaptive weight, $G_L$ is the last decoder layer, $\delta = 10^{-6}$, and $\mathcal{L}_{LPIPS}$ is the exact distance metric described in Zhang et al. (2012).

## 2.2 MASKGIT

MaskGit (Chang et al., 2022) is a generative model that models distributions over tokens, such as produced by a VQ-GAN. Instead of autoregressively modelling the sequence of tokens, MaskGit generates images with competitive sample quality at a fraction of the sampling cost by using a masked token prediction objective during training. Formally, we denote $z \in \mathbb{Z}^{H \times W}$ as the discrete latent tokens representing an image. For each training step, we uniformly sample $t \in [0, 1)$ and randomly generate a mask $m \in \{0, 1\}^{H \times W}$ with $N = \lceil \gamma H W \rceil$ masked values, where $\gamma = \cos\left(\frac{\pi}{2} t\right)$. Then, MaskGit learns to predict the masked tokens with the following objective

$$\mathcal{L}_{\text{mask}} = -\operatorname{E}_{z \in \mathcal{D}} \left[ \log p(z \mid z \odot m) \right]. \tag{4}$$

During inference, because MaskGit has been trained to model any set of unconditional and conditional probabilities, we can sample any subset of tokens per sampling iteration, from the extreme case of sampling all tokens (independent) to sampling one token at a time (autoregressive). Chang et al. (2022) introduces a confidence-based sampling mechanism whereas other work (Lee et al., 2022) proposes iterative sample-and-revise approaches.

## 3 TECO

Generating temporally consistent videos requires training on long videos to correctly learn long-term temporal dependencies between frames. However, computational and memory requirements remain the primary bottleneck in preventing from doing so. We present **Te**mporally **Co**nsistent Video Transformer (TECO), a video generation model that more efficiently scales to training on longer horizon videos.

First, we train a VQ-GAN to spatially compress our video data. Shown in prior work (Seo et al., 2022), this is an important step for video prediction in a more efficient and scalable manner. However, even in latent space, existing methods are still limited to modeling short sequences of 16–24 frames, which can be attributed to the quadratics costs of transformer layers as sequence length grows. With 256 tokens per frame, 16 frame videos already consist of 4096 tokens, and scaling to longer videos of 100s frames is prohibitively expensive, where resulting videos have tens of thousands of tokens. Therefore, in the following sections, we propose several key design choices to building a more efficient video prediction model.

### 3.1 VECTOR-QUANTIZED LATENT DYNAMICS

Our proposed framework shown in Figure 2 follows similarly to prior work in latent dynamics models (Hafner et al., 2019; 2020; Saxena et al., 2021), with several key differences in architectural and latent variable design. Let $x_{1:T}$ consist of a sequence of video frames encoded using a pretrained

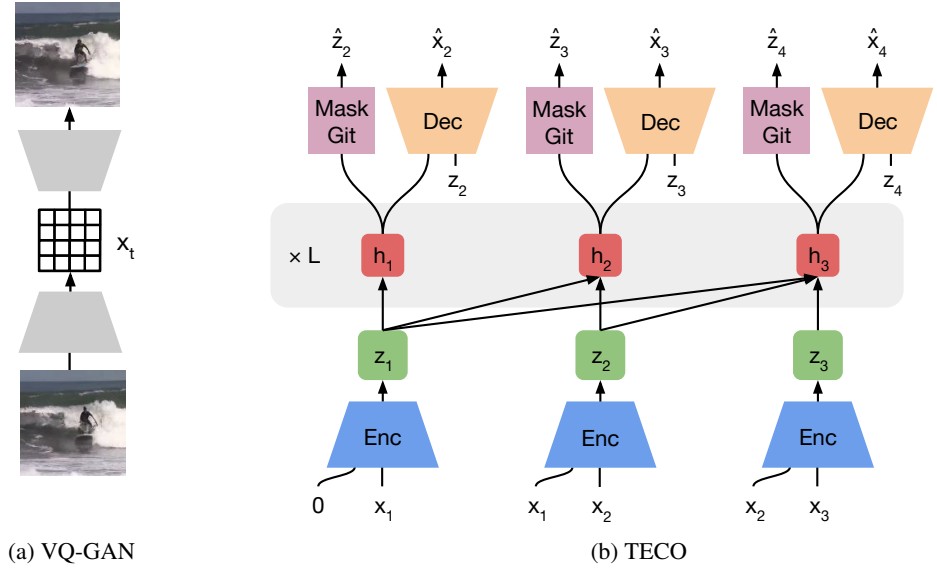

(a) VQ-GAN             (b) TECO

Figure 2: The architectural design of TECO. Our proposed method models sequences of videos encoded with a pretrained VQ-GAN. We achieve efficient and scalable training and generation on long sequences through several key design choices to maximally compress our representations. We leverage temporal redundancies by encoding frames conditioned on the previous one, and model temporal dependencies in a downsampled latent space. For fast sampling, we learn a MaskGit dynamics for the prior.

**VQ-GAN.** In the following sections, we motivate each component for our model, with several specific design choices to ensure efficiency and scalability. TECO consists of four components:

$$
\begin{array}{llll}
\text{Encoder:} & z_t = E(x_t, x_{t-1}) & \text{Temporal Transformer:} & h_t = H(z_{\leq t}) \\
\text{Dynamics Prior:} & p(z_t \mid h_{t-1}) & \text{Decoder:} & p(x_t \mid z_t, h_{t-1})
\end{array}
\tag{5}
$$

**Encoder** Although VQ-GAN exploits spatial redundancies, we can achieve more compressed representations by leveraging temporal redundancy in video data. To do this, we learn a CNN encoder $z_t = E(x_t, x_{t-1})$ which encodes the current frame $x_t$ conditioned on the previous frame by channel-wise concatenating $x_{t-1}$, and then quantizes the output using codebook $C$ to produce $z_t$. We apply the VQ loss defined in Equation (1) per timestep. In addition, we $\ell_2$-normalize the codebook and embeddings to encourage higher codebook usage (Yu et al., 2021). Conditionally encoding nearby frames lets the model learn smaller latents, and provides a general way to take advantage of temporal redundancy. The most common form of temporal redundancy is the large amount of shared bits between neighboring frames, generally only differing in small movements, such as slight camera shifts, or objects moving slightly. The first frame is concatenated with zeros and does not quantize $z_1$ to prevent information loss. As we focus on video prediction, there is always at least 1 frame to condition on, so we do not need to predict the un-quantized representation of the first frame when computing decoding and dynamics losses. Intuitively, this also does not burden the dynamics model to learn an unconditional prior.

**Temporal Transformer** Compressed, discrete latents are more lossy and tend to require higher spatial resolutions compared to continuous latents. Therefore, before modeling temporal information, we apply a single strided convolution to downsample each discrete latent $z_t$, where visually simpler datasets allow for more downsampling and visually complex datasets require less downsampling. Afterwards, we learn a large transformer to model temporal dependencies, and then apply a transposed convolution to upsample our representation back to the original resolution of $z_t$. In summary, we use the following architecture:

$$
h_t = H(z_{<t}) = \text{ConvTranspose}(\text{Transformer}(\text{Conv}(z_{<t})))
\tag{6}
$$

**Decoder** The decoder is an upsampling CNN that reconstructs $\hat{x}_t = D(z_t, h_t)$, where $z_t$ can be interpreted as the posterior of timestep $t$, and $h_t$ is the output of the temporal transformer which summarizes information from previous timesteps. $z_t$ and $h_t$ are concatenated channel-wise before

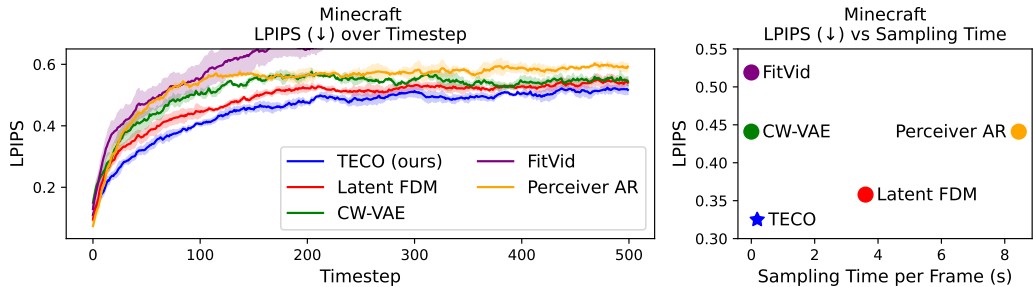

Figure 3: Quantitative comparisons between TECO and baseline methods in long-horizon temporal consistency (left) and sampling speed (right). Our method is able to remain temporally consistent while still generating sharp samples with fast sampling speed.

being fed into the decoder. Together with the encoder, the decoder optimizes the following cross entropy reconstruction loss

$$\mathcal{L}_{\text{recon}} = -\frac{1}{T} \sum_{t=1}^{T} \log p(x_t \mid z_t, h_t).\qquad(7)$$

which encourages $z_t$ features to encode relative information between frames since the temporal transformer can aggregate information over time. This allows us to learn more compressed codes that enable more efficient modeling over longer sequences.

**Dynamics Prior**  Lastly, we use a MaskGit (Chang et al., 2022) to model the dynamics prior, $p(z_t \mid h_t)$. In our experiments, we show that using a MaskGit prior allows for not just faster but also higher quality sampling compared to an autoregressive prior. During every training iteration, we use the same process as prior work to sample a random mask $m_t$ and optimize

$$\mathcal{L}_{\text{prior}} = -\frac{1}{T} \sum_{t=1}^{T} \log p(z_t \mid z_t \odot m_t).\qquad(8)$$

where $h_t$ is concatenated channel-wise with masked $z_t$ to predict the masked tokens. During generation, we follow Lee et al. (2022), where we initially generate each frame in chunks of 8 at a time and then go through 2 revise rounds of re-generating half the tokens each time.

**Training Objective**  The final objective is the sum of these losses:

$$\mathcal{L}_{\text{TECO}} = \mathcal{L}_{\text{VQ}} + \mathcal{L}_{\text{recon}} + \mathcal{L}_{\text{prior}}\qquad(9)$$

## 3.2  DropLoss

To train the model efficiently on long videos, we propose DropLoss, a simple trick to allow for more scalable and efficient training. Due to its architecture design, TECO can be separated into two components: (1) learning temporal representations, consisting of the encoder and the temporal transformer, and (2) predicting future frames, consisting of the dynamics prior and decoder. We can increase training efficiency by dropping out random timesteps that are not decoded and thus omitted from the reconstruction loss. For example, given a video of T frames, we compute $h_t$ for all $t \in \{1, \ldots, T\}$, and then compute the losses $\mathcal{L}_{\text{prior}}$ and $\mathcal{L}_{\text{recon}}$ for only 10% of the indices. Because random indices are selected each iteration, the model still needs to learn to accurately predict all timesteps. This reduces training costs significantly because the decoder and dynamics prior require non-trivial computations. DropLoss is applicable to both a wide class of architectures and to tasks beyond video prediction.

## 4  Experiments

### 4.1  Datasets

We introduce three challenging video datasets to better measure long-range consistency in video prediction. We design these benchmarks around 3D environments in DMLab (Beattie et al., 2016), Minecraft (Guss et al., 2019), and Habitat (Savva et al., 2019), with videos of agents randomly traversing different scenes of varying difficulty. These datasets require video prediction models to re-produce observed parts of scenes, and newly generate unobserved parts of the scene. In contrast,

Table 1: Quantitative evaluation on all four datasets. Detailed results in Appendix K.

| Method | DMLab | | | | Minecraft | | | |
|---|---|---|---|---|---|---|---|---|
| | FVD ↓ | PSNR ↑ | SSIM ↑ | LPIPS ↓ | FVD ↓ | PSNR ↑ | SSIM ↑ | LPIPS ↓ |
| FitVid | 176 | 12.0 | 0.356 | 0.491 | 956 | 13.0 | 0.343 | 0.519 |
| CW-VAE | 125 | 12.6 | 0.372 | 0.465 | 397 | 13.4 | 0.338 | 0.441 |
| Perceiver AR | 96 | 11.2 | 0.304 | 0.487 | **76** | 13.2 | 0.323 | 0.441 |
| Latent FDM | 181 | 17.8 | 0.588 | 0.222 | 167 | 13.4 | 0.349 | 0.429 |
| TECO (ours) | **48** | **21.9** | **0.703** | **0.157** | 116 | **15.4** | **0.381** | **0.340** |

| Method | Habitat | | | | Kinetics-600 | | | |
|---|---|---|---|---|---|---|---|---|
| | FVD ↓ | PSNR ↑ | SSIM ↑ | LPIPS ↓ | FVD ↓ | PSNR ↑ | SSIM ↑ | LPIPS ↓ |
| Perceiver AR | 164 | **12.8** | **0.405** | 0.676 | 1022 | 13.4 | 0.310 | 0.404 |
| Latent FDM | 433 | 12.5 | 0.311 | **0.582** | 960 | 13.2 | 0.334 | 0.413 |
| TECO (ours) | **73** | **12.8** | 0.363 | 0.604 | **799** | **13.8** | **0.341** | **0.381** |

many existing video benchmarks do not have strong long-range dependencies, where a model with limited context is sufficient. Refer to Appendix N for further details on each dataset.

**DMLab**  DeepMind Lab is a simulator that procedurally generates random 3D mazes with random floor and wall textures. We generate 40k action-conditioned $64 \times 64$ videos of 300 frames of an agent randomly traversing $7 \times 7$ mazes by choosing random points in the maze and navigating to them via the shortest path. We train all models for both action-conditioned and unconditional prediction (by periodically masking out actions) to enable both types of generations. We use both modes to evaluate since a video model may generate new parts of a scene that do not correlate with the action (e.g. run into a wall) which results in out-of-distribution errors. However, action-conditioning is useful with enough conditioned past context, and substantially lowers variance on PSNR, SSIM, and LPIPS evaluations.

**Minecraft**  This popular game features procedurally generated 3D worlds that contain complex terrain such as hills, forests, rivers, and lakes. We collect 200k action-conditioned videos of length 300 and resolution $128 \times 128$ in Minecraft's marsh biome. The player iterates between walking forward for a random number of steps and randomly rotating left or right, resulting in parts of the scene going out of view and coming back into view later. We train action-conditioned for all models for ease of interpreting and evaluating, though it is generally easy for video models to unconditionally learn these discrete actions.

**Habitat**  Habitat is a simulator for rendering trajectories through scans of real 3D scenes. We compile ∼1400 indoor scans from HM3D (Ramakrishnan et al., 2021), MatterPort3D (Chang et al., 2017), and Gibson (Xia et al., 2018) to generate 200k action-conditioned videos of 300 frames at a resolution of $128 \times 128$ pixels. We use Habitat's in-built path traversal algorithm to construct action trajectories that move our agent between randomly sampled locations. Similar to DMLab, we train all video models to perform both unconditional and action-conditioned prediction.

**Kinetics-600**  Kinetics-600 (Carreira & Zisserman, 2017) is a highly complex real-world video dataset, originally proposed for action recognition. The dataset contains ∼400k videos of varying length of up to 300 frames. We evaluate our method in the video prediction without actions (as they do not exist), generating 80 future frames conditioned on 20. In addition, we filter out videos shorter than 100 frames, leaving 392k videos that are split for training and evaluation. We use a resolution of $128 \times 128$ pixels. Although Kinetics-600 does not have many long-range dependencies, we evaluate our method on this dataset to show that it can scale to complex, natural video.

## 4.2 BASELINES

We compare against SOTA baselines selected from several different families of models: latent-variable-based variational models, autoregressive likelihood models, and diffusion models. In addition, for more fair comparisons, we train all models on VQ codes using the same VQ-GAN as our method. For our diffusion baseline, we follow Rombach et al. (2022) and use a pretrained VAE instead of a VQ-GAN. Note that we do not have any GANs for our baselines, since to the best

of our knowledge, there does not exist a GAN that trains on latent space instead of raw pixels, an important aspect for properly scaling to long video sequences.

**FitVid**  FitVid (Babaeizadeh et al., 2021) is a state-of-the-art variational video prediction model based on CNNs and LSTMs that scales to complex video by leveraging efficient architectural design choices in its encoder and decoder.

**Clockwork VAE**  CW-VAE (Saxena et al., 2021) is also a variational video prediction model that is designed to better learn long-range dependencies through a hierarchies of latent variables with exponentially slower tick speeds for each new level.

**Perceiver AR**  We use Perceiver AR (Hawthorne et al., 2022) as our AR baseline over VQ-GAN discrete latents, which has been show to be an effective generative model that can efficiently incorporate long-range sequential dependencies. Conceptually, this baseline is similar to HARP (Seo et al., 2022) with a Perceiver AR as the prior instead of a sparse transformer (Child et al., 2019). We choose Perceiver AR over other autoregressive baselines such as VideoGPT (Yan et al., 2021) or TATS (Ge et al., 2022) primarily due to the prohibitive costs of transformers when applied to tens of thousands of tokens.

**Latent FDM**  For our diffusion baseline, we train a Latent FDM model with frame-wise autoregressive sampling.  Although FDM (Harvey et al., 2022) is originally trained on pixel observations, we also train in latent space for a more fair comparison with our method and other baselines, as training on long sequences in pixel space is too expensive. We follow LDM (Rombach et al., 2022) to separately train an autoencoder to encode each frame into a set of continuous latents.

### 4.3 EXPERIMENTAL SETUP

**Training**  All of our models are trained for 1 million iterations under fixed compute budget (measured in TPU v3 days) allocated for each dataset.  Models are trained on TPU-v3 instances, ranging from v3-8 to v3-128 TPU pods (similar to 4 V100s to 64 V100s) with training times of roughly 3-5 days.  For DMLab, Minecraft, and Habitat we train all models on full 300 frames videos, and 100 frames for Kinetics-600.  Our VQ-GANs are trained on 8 A5000 GPUs, taking about 2-4 days for each dataset, and downsample all videos to $16 \times 16$ grids of discrete latents per frame regardless of original video resolution.  More details on exact hyperparameters and compute budgets for each dataset can be found in Appendix O.

**Evaluation**  We evaluate our models using a combination of standard video prediction metrics such as PSNR (Huynh-Thu & Ghanbari, 2008), SSIM (Wang et al., 2004), LPIPS (Zhang et al., 2012), and FVD (Unterthiner et al., 2019).  For DMLab, Minecraft, and Habitat, we measure FVD on 300 frame videos, conditioned on 36 frames (264 predicted frames). For Kinetics-600, we evaluate FVD on 100 frame videos, conditioned on 20 frames (80 predicted frames). To evaluate temporal consistency, we measure PSNR, SSIM, and LPIPS on video predictions conditioned on 144 frames (156 predicted frames), and action condition for all models.  Conditioning on a large portion of the video ensures that the model can observe a large part of the scene, and combined with action-conditioning, the model with temporally-consistent predictions should generate future frames close to the ground truth. Due to this reduced stochasticity, we only sample one prediction for computing PSNR, SSIM, and LPIPS. We compute all metrics over batches of 256 examples, averaged over 4 runs to make 1024 total samples.

### 4.4 BENCHMARK RESULTS

**DMLab & Minecraft**  Table 1 shows quantitative results on the DMLab and Minecraft datasets. TECO performs the best across all metrics for both datasets when training on the full 300 frame videos. Figure 4 shows sample trajectories and 3D visualizations of the generated DMLab mazes, where TECO is able to generate more stable and consistent 3D mazes. For both datasets, CW-VAE, FitVid, and Perceiver AR can produce sharp predictions, but do not model long-horizon context well, with per-frame metrics sharply dropping as the future prediction horizon increases as seen in Figure C.1. Latent FDM has consistent predictions, but high FVD most likely due to FVD being sensitive to high frequency errors.

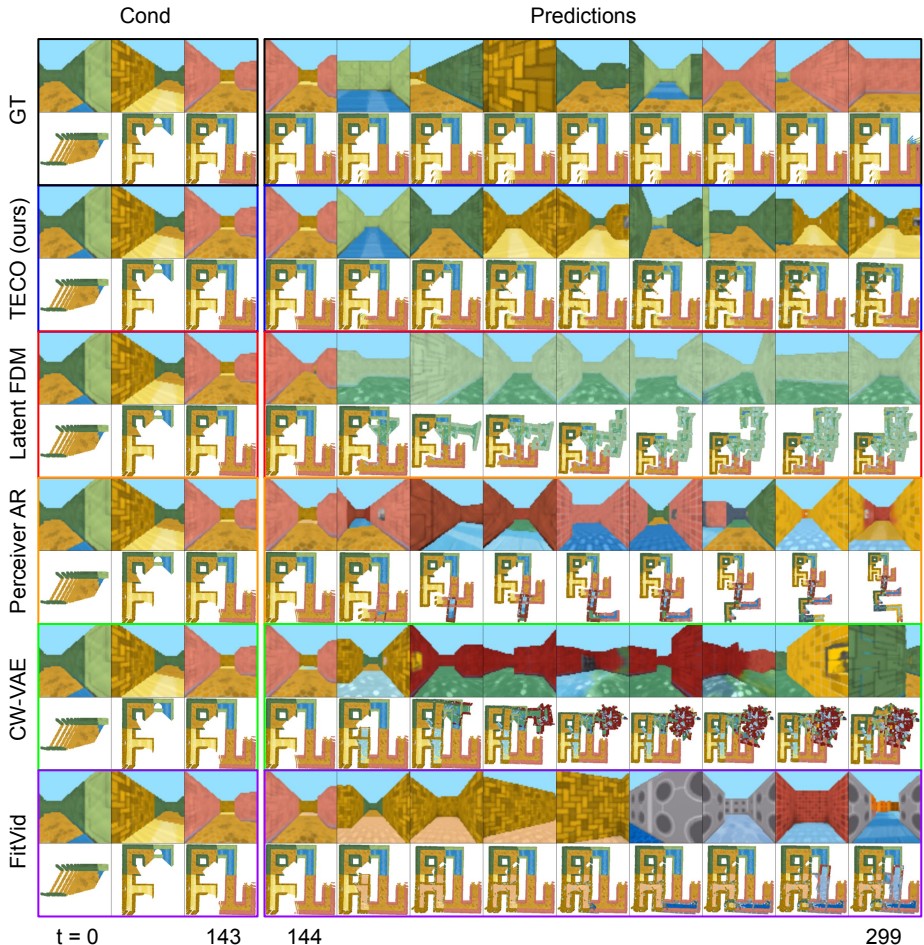

Figure 4: 3D visualization of predicted trajectories in DMLab for each model, generating 156 frames conditioned on 144. TECO is the only model that retain maze consistency with ground-truth, whereas baselines tend to extend out of the maze or create fictitious corridors that did not exist. **Video predictions use only the first-person RGB frames**. Refer to Appendix N.1 for more details on 3D evaluation. A video corresponding to this figure is available at: https://sites.google.com/view/iclr23-teco.

In order to better investigate scaling properties of our models, Figure D.1 and Figure D.2 compare TECO and Latent FDM on different training sequence lengths. Intuitively, under a fixed computation budget and batch size, models that train on shorter sequence lengths can scale larger, with more FLOPs allocated per frame. In general, this is reflected in model architectures through computations at higher spatial resolutions (e.g. less downsampling). For DMLab, we see that in terms of per-frame metrics, models generally benefit from training on longer videos, where more computation per image has less of an effect due to saturation in image quality because of relatively simple visual complexity. For Minecraft, we observe that models generally perform best when training with 100 frames of context, which have better per-image sample quality compared to training on 300 frames due to higher downsampling required for longer sequences. Models trained on 300 frames generally have more distortion in predictions compared to 100 frames. Theoretically, as the compute budget is increased, training on 300 frames would eventually outperform models trained on 100 frames.

**Habitat** Table 1 shows results for our Habitat dataset. We only evaluate our strongest baselines, Perceiver AR and Latent FDM due to the need to implement model parallelism. Because of high complexity of Habitat videos, all models generally perform equally as bad in per-frame metrics. However, TECO has significantly better FVD. Qualitatively, Latent FDM quickly collapses to blurred predictions with poor sample quality, and Perceiver AR can generate high quality frames, though less temporally consistent than TECO: agents in Habitat videos navigate to far points in the scene and back whereas Perceiver AR tends to generate samples where the agent is constantly turning. TECO generates traversals of a scene that match the data distribution more closely.

**Kinetics-600** Table 1 shows FVD for predicting 80 $128 \times 128$ frames conditioned on 20 for Kinetics-600. Although Kinetics-600 does not have many long-range dependencies, we found that TECO is able to produce more stable generations that degrade slower by incorporating longer contexts. In contrast, Perceiver AR tends to degrade quickly, with Latent FDM performing in between. Figure K.1 and Table K.4 include further investigations using top-k sampling for Perceiver AR and TECO. Table 1 does not use top-k sampling for a fair comparison against Latent FDM. With top-k sampling, Perceiver AR outperforms our method at $k = 8$. However, resulting videos tend to be uninteresting with little to no dynamics movement.

**Sampling Speed** Figure 3 compares sampling speed for all models. We report sampling speed on Minecraft and observed similar results for the different model sizes used on other datasets. FitVid and CW-VAE are both significantly faster that other methods, but have poor sample quality. On the other end, Perceiver AR and Latent FDM can produce high quality samples, but are 20-60x slower than TECO, which has comparably fast sampling speed while retaining high sample quality.

### 4.5 ABLATIONS

In this section, we perform ablations on various architectural decisions of our model. For simplicity, we evaluate our methods on short sequences of 16 frames from Something-Something-v2 (SSv2). We choose SSv2 as it provides insight into scaling our method on complex real-world data more similar to Kinetics-600 while being computationally cheaper to run.

Table F.1 shows several ablations comparing posterior, prior, and various architectural design choices. We demonstrate that using VQ-latent dynamics with a MaskGit prior proves better compared to alternative formulations for latent dynamics models, such as popular variational methods. In addition, we show that conditional encodings learn better representations for video predictions. We also ablate the codebook size, showing that although there exists an optimal codebook size, it does not matter too much as along as there are not too many codes, which may make it more difficult for the prior to learn. Lastly, we show the benefits of DropLoss, with up to 60% faster training and a minimal increase in FVD. The benefits are greater for longer sequences, and allow video models to better account for long horizon context with little cost in performance.

Table F.2 shows ablations on scaling different parts of our model, such as the encoder, decoder, temporal transformer, and prior. In general, it is more beneficial to have an imbalanced encoder decoder architecture, with more parameters in the decoder. For the temporal transformer, it is more beneficial to have larger resolution features ($4 \times 4$), especially for more complex data like SSv2, and less useful for visually simpler datasets such as DMLab or Minecraft. Similarly, having a larger width is more beneficial than more layers due to increased capacity to represent each frame. Lastly, for scaling the MaskGit prior, more layers is better than larger width networks.

## 5 DISCUSSION

We introduced TECO, an efficient video prediction model that leverages hundreds of frames of temporal context. Our evaluation demonstrated that TECO accurately incorporates long-range context, outperforming SOTA baselines across a wide range of datasets. In addition, we introduce several difficult video datasets, which we hope make it easier to evaluate temporal consistency in future video prediction models. We identify several limitations as directions for future work:

- Although we show that PSNR, SSIM, and LPIPS can be reliable metrics to measure consistency when video models are properly conditioned, there remains room for better evaluation metrics that provide a reliable signal as the prediction horizon grows, since new parts of a scene that are generated are unlikely to correlate with ground truth.

- Our focus was on learning a compressed tokens and an expressive prior, which we combined with a simple full attention transformer as the sequence model. Leveraging prior work on efficient sequence models (Choromanski et al., 2020; Wang et al., 2020; Zhai et al., 2021; Gu et al., 2021; Hawthorne et al., 2022) would likely allow for further scaling.

- We trained all models on top of pretrained VQ-GAN codes to reduce the data dimensionality. This compression step lets us train on longer sequences at a cost of reconstruction error, which causes noticeable artifacts in Kinetics-600, such as corrupted text and incoherent faces. Although TECO can train directly on pixels, a $\ell_2$ loss results in slightly blurry predictions. Training directly on pixels with diffusion or GAN losses would be promising.

## 6 REPRODUCIBILITY

We provide several resources in order to aim for better reproducilbity. We include anonymized code in the supplementary materials for our models, baselines, and datasets. In addition, Appendix O details hyperparameters and compute requirements for all models.

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

# A   SAMPLING PROCESS

Given a sequence of conditioning frames, $o_1, \ldots, o_t$, we encode each frame using the pretrained VQ-GAN to produce $x_1, \ldots, x_t$, and then use the conditional encoder to compute $z_1, \ldots, z_t$. In order to generate the next frame, we use the temporal transformer to compute $h_t$, and feed it into the MaskGit dynamics prior to predict $\hat{z}_{t+1}$. Let $z_{t+1} = \hat{z}_{t+1}$ and feed it through the temporal tranformer and MaskGit to predict $\hat{z}_{t+2}$. We repeat this process until the entire trajectory is predicted, $\hat{z}_{t+1}, \ldots, \hat{z}_T$. In order to decode back into frames, we first decode into the VQ-GAN latents, and then decode to RGB using the VQ-GAN decoder. Note that generation can be completely done in latent space, and rendering back to RGB can be done in parallel over time once the latents for all timesteps are computed.

# B SAMPLES

## B.1 DMLAB

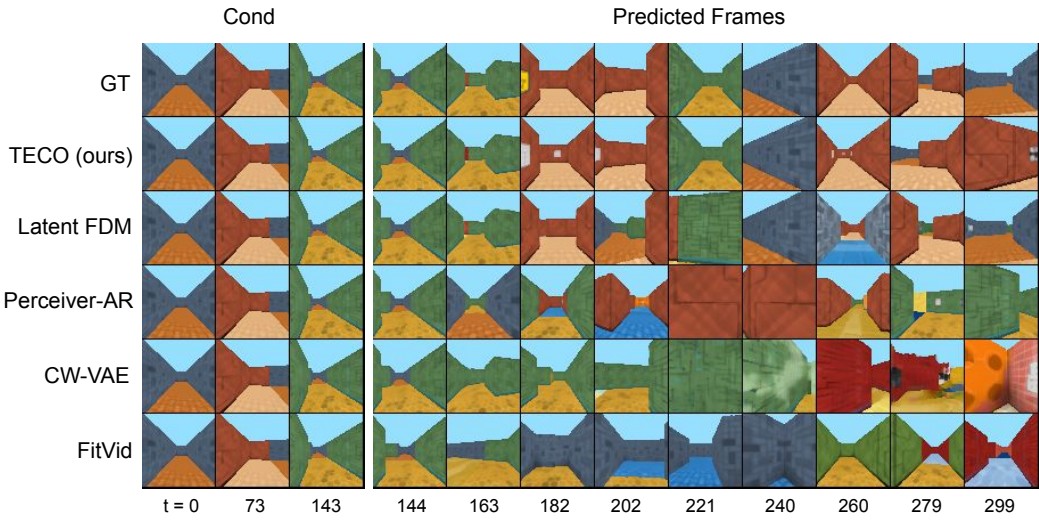

Figure B.1: 156 frames generated conditioned on 144 (action-conditioned)

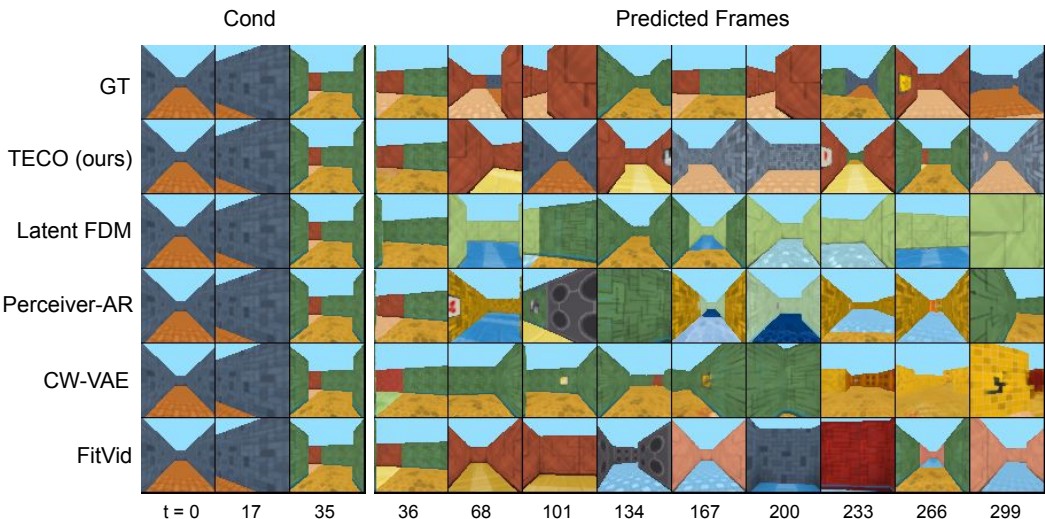

Figure B.2: 264 frames generated conditioned on 36 (**no** action-conditioning)

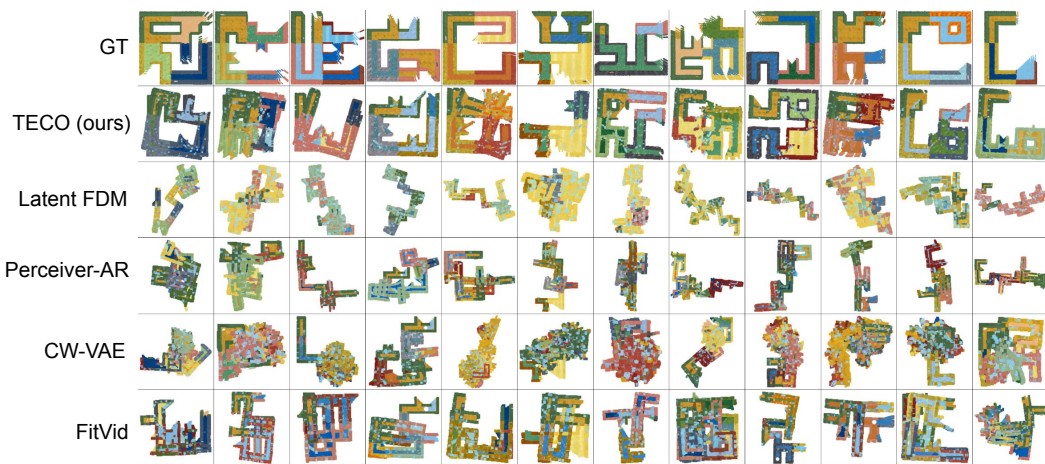

Figure B.3: 3D visualizations of the resulting generated DMLab mazes

## B.2 MINECRAFT

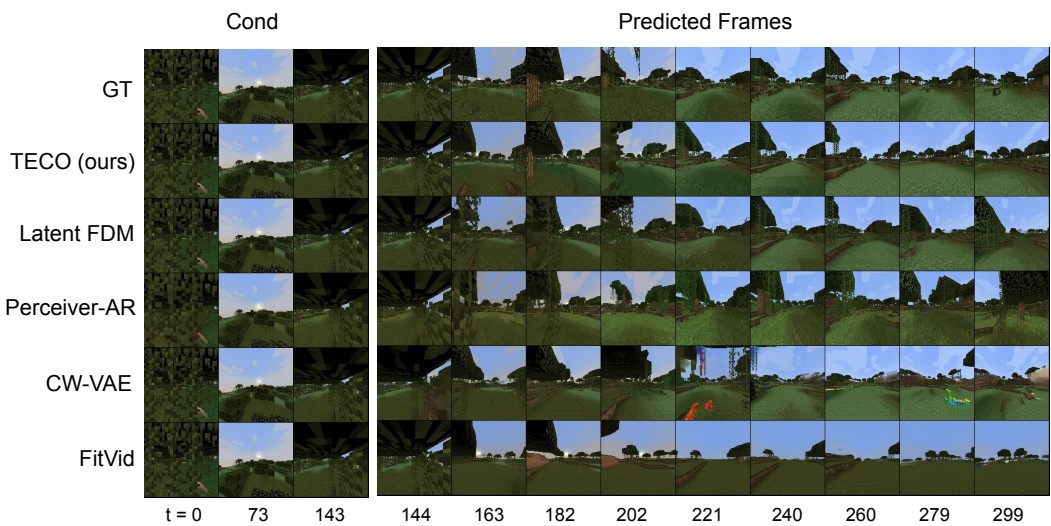

Figure B.4: 156 frames generated conditioned on 144 (action-conditioned)

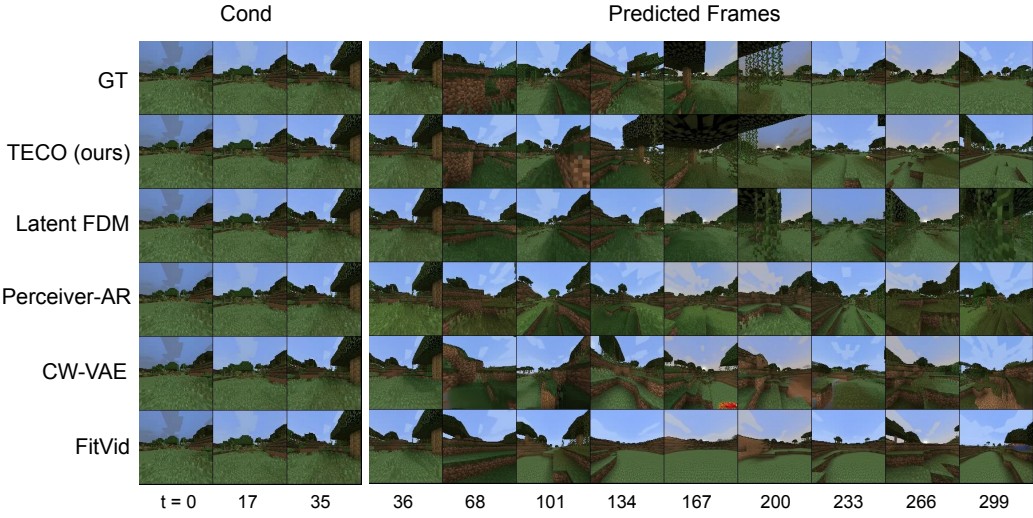

Figure B.5: 264 frames generated conditioned on 36 (action-conditioned)

## B.3 HABITAT

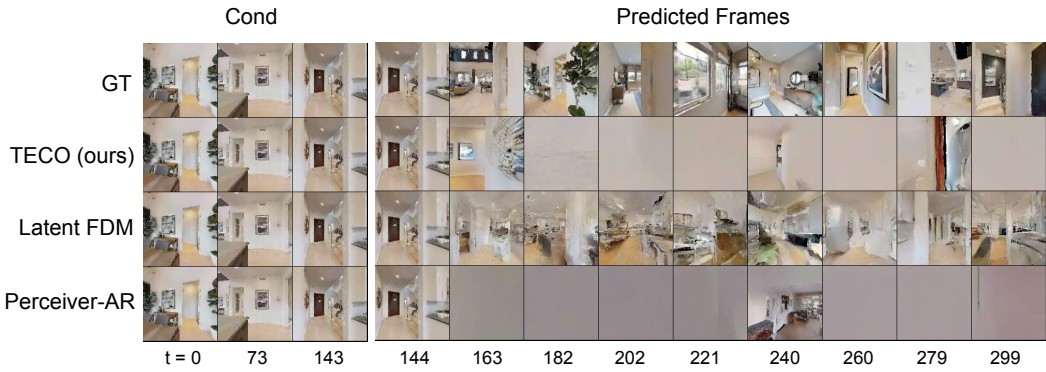

Figure B.6: 156 frames generated conditioned on 144 (action-conditioned)

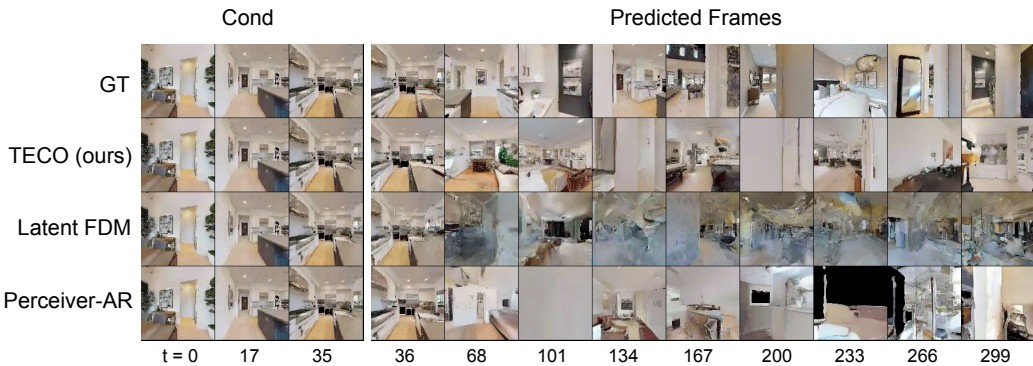

Figure B.7: 264 frames generated conditioned on 36 (**no** action-conditioning)

## B.4 KINETICS-600

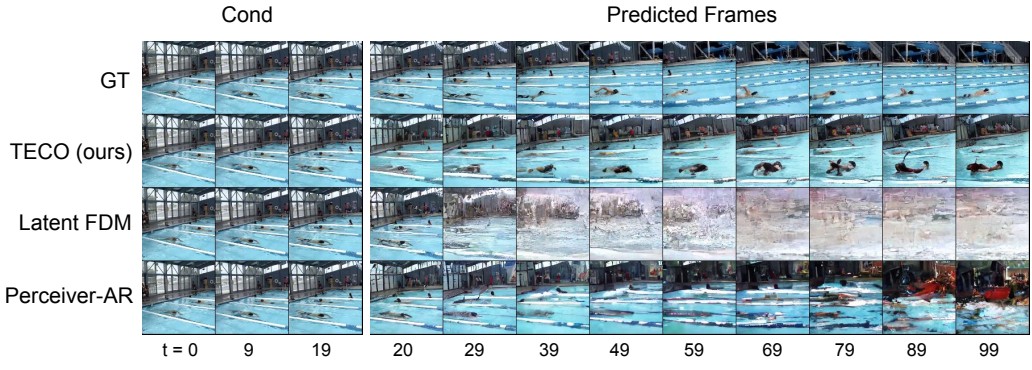

Figure B.8: 80 frames generated conditioned on 20 (no top-k sampling)

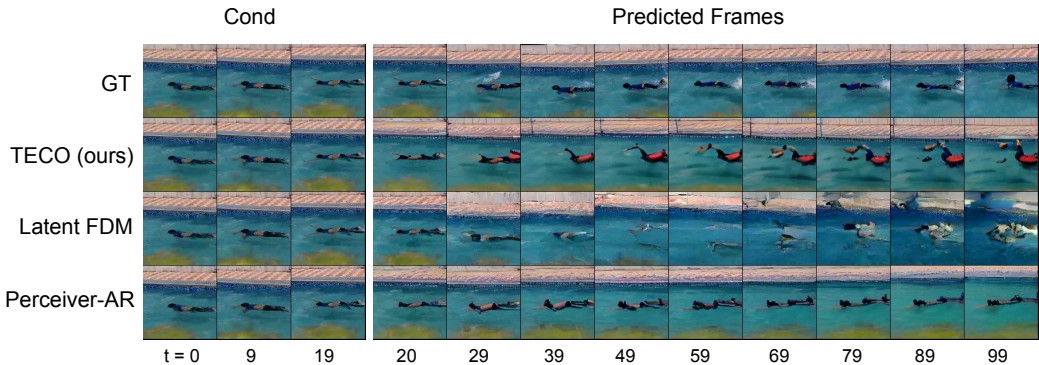

Figure B.9: 80 frames generated conditioned on 20 (with top-k sampling)

## C PERFORMANCE VERSUS HORIZON

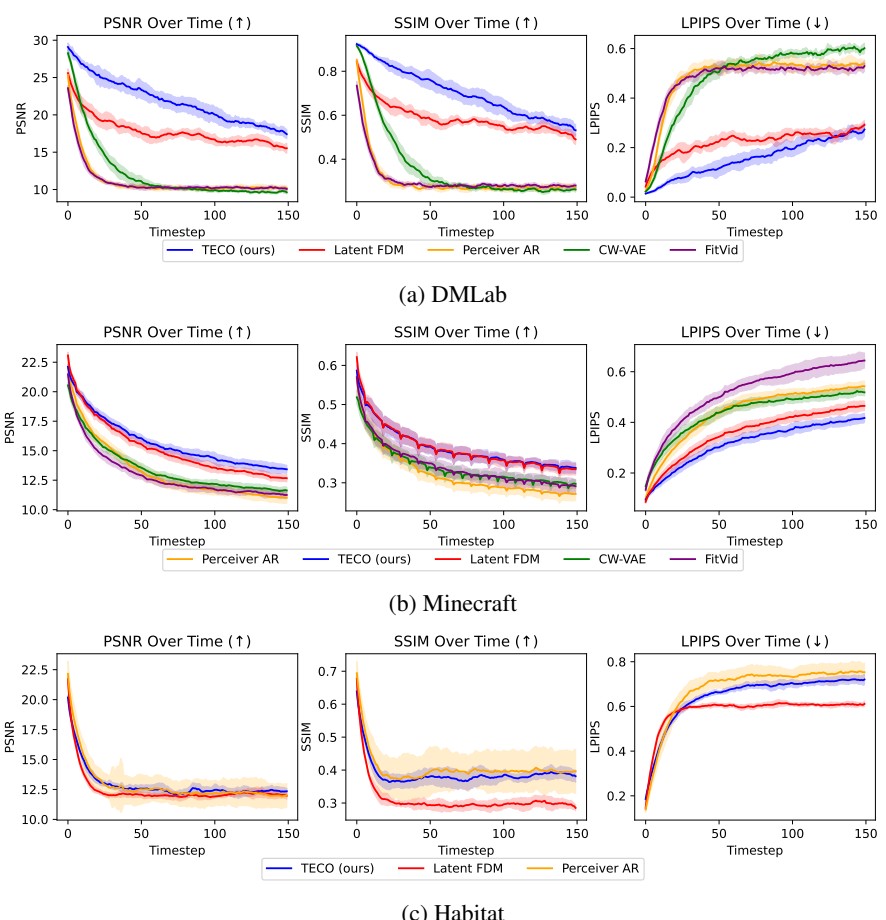

(a) DMLab

(b) Minecraft

(c) Habitat

Figure C.1: All plots shows PSNR, SSIM, and LPIPS on 150 predicted frames conditioned on 144 frames. The 144 conditioned frames are not shown on the graphs and timestep 0 corresponds to the first predicted frame

Figure C.1 shows PSNR, SSIM, and LPIPS as a function of prediction horizon for each dataset. Generally, each plot reflected the corresponding aggregated metrics in Table 1. For DMLab, TECO shows much better temporal consistency for the full trajectory, with Latent FDM coming in second. CW-VAE is able retain some consistency but drops fairly quickly. Lastly, FitVid and Perceiver AR lose consistency very quickly. We see a similar trend in Minecraft, with Latent FDM coming closer in matching TECO. For Habitat, all methods generally have trouble producing consistent predictions, primarily due to the difficulty of the environment.

# D PERFORMANCE VERSUS TRAINING SEQUENCE LENGTH

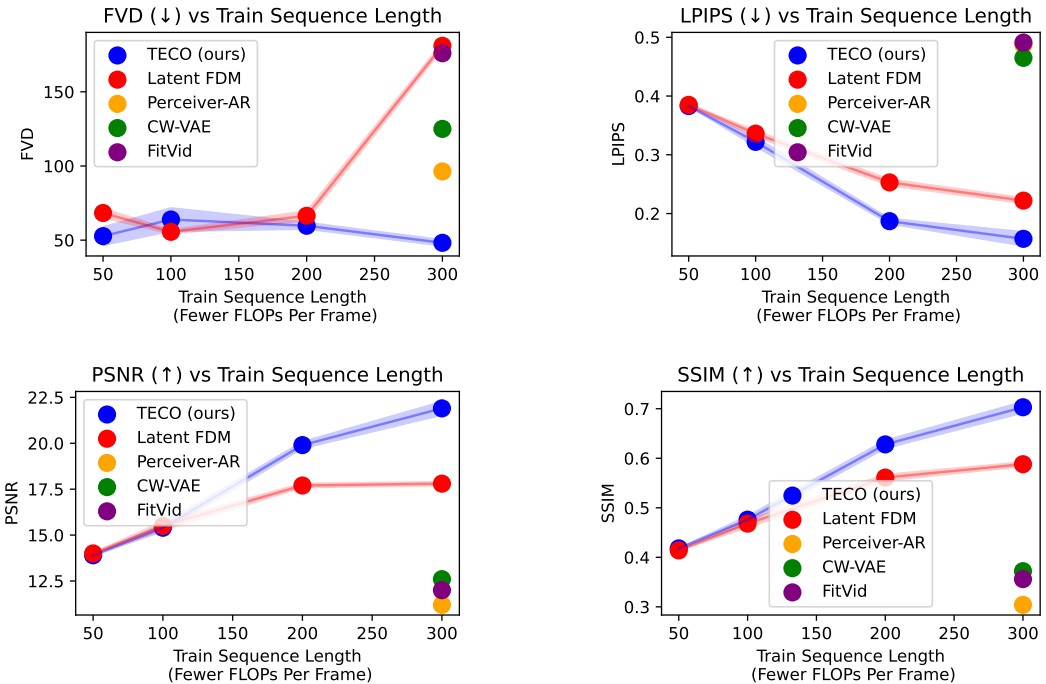

Figure D.1: DMLab

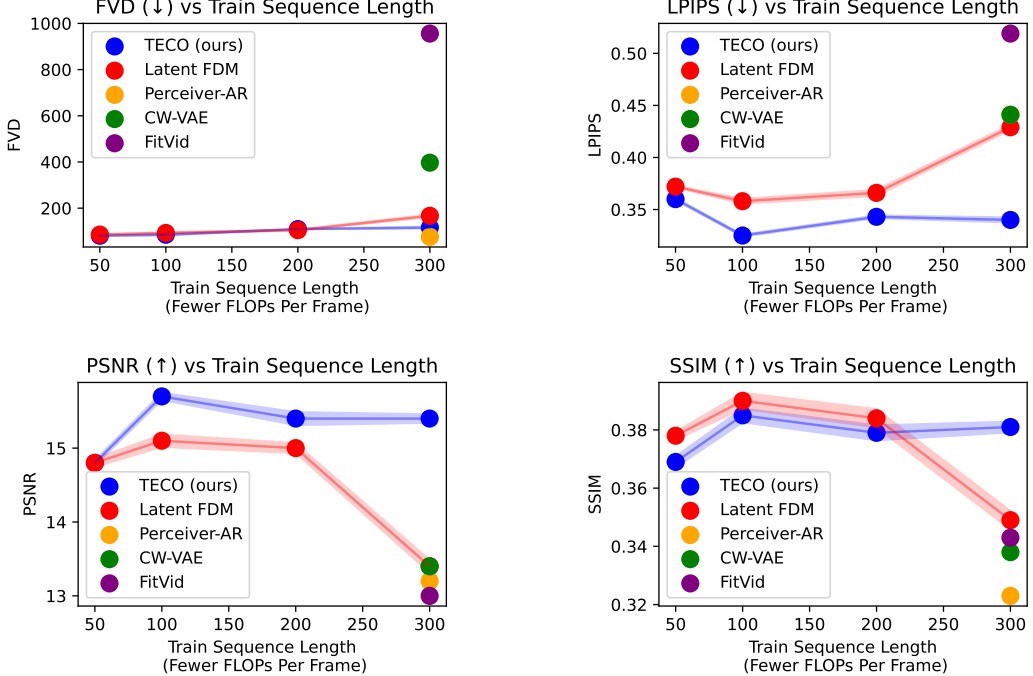

Figure D.2: Minecraft

Figure D.1 and Figure D.2 show plots comparing performance with training models on different sequence lengths. Under a fixed compute budget and batch size, training on shorter videos enables us to scale to larger models. This can also be interpreted as model capacity or FLOPs allocated per image. In general, training on shorter videos enables higher quality frames (per-image) but at a cost of worse temporal consistency due to reduced context length. We can see a very clear trend in DMLab, in that TECO is able to better scale on longer sequences, and correspondingly benefits from it. Latent FDM has trouble when training on full sequences. We hypothesize that this may be due to diffusion models being less amenable towards downsamples, it it needs to model and predict noise. In Minecraft, we see the best performance at around 50-100 training frames, where a model has higher fidelity image predictions, and also has sufficient context.

## E  SAMPLING

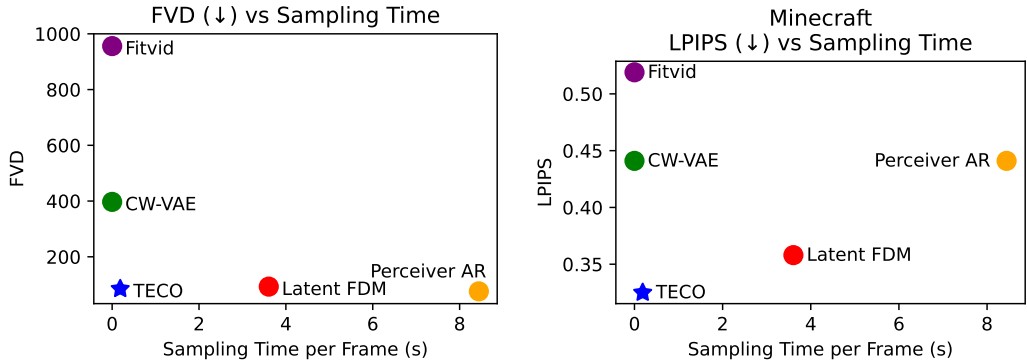

| | Sampling Time per Frame (ms) |
|---|---|
| TECO (ours) | 186 |
| Latent FDM | 3606 |
| Perceiver-AR | 8443 |
| CW-VAE | 0.062 |
| FitVid | 0.074 |

## F  ABLATIONS

| DropLoss Rate | FVD | Train Step (ms) |
|---|---|---|
| 0.8 | 187 | 125 |
| 0.6 | 186 | 143 |
| 0.4 | 184 | 155 |
| 0.2 | 184 | 167 |
| 0.0 | 182 | 182 |

(a) DropLoss Rates

| Posteriors | FVD |
|---|---|
| VQ (+ MaskGit prior) (ours) | 189 |
| OneHot (+ MaskGit prior) | 199 |
| OneHot (+ Block AR prior) | 209 |
| OneHot (+ Independent prior) | 228 |
| Argmax (+ MaskGit prior) | 336 |

(b) Posteriors

| Dynamics Prior | FVD |
|---|---|
| MaskGit (ours) | 189 |
| Independent | 220 |
| Autoregressive | 207 |

(c) Prior Networks

| Conditional Encoding | FVD |
|---|---|
| Yes (ours) | 189 |
| No | 208 |

(d) Conditional Encoding

| Number of Codes | FVD |
|---|---|
| 64 | 191 |
| 256 | 195 |
| 1024 | 186 |
| 4096 | 200 |

(e) VQ Codebook Size

Table F.1: Ablations comparing alternative prior, posterior, and codebook designs

| Size | FVD 2 × 2 | 4 × 4 | Layers | Width | FVD 2 × 2 | 4 × 4 | Layers | Width | FVD 2 × 2 | 4 × 4 |
|---|---|---|---|---|---|---|---|---|---|---|
| Base | 204 | 189 | 8 | 768 | 204 | 189 | 8 | 768 | 204 | 189 |
| Small Enc | 214 | 191 | 8 | 384 | 260 | 196 | 8 | 384 | 228 | 193 |
| Small Dec | 232 | 198 | 2 | 768 | 216 | 202 | 2 | 768 | 228 | 201 |

| (a) Encoder and Decoder | (b) Temporal Transformer | (c) MaskGit Prior |
|---|---|---|

Table F.2: Ablations on scaling different parts of TECO.

| | FVD ($\downarrow$) | PSNR ($\uparrow$) | SSIM ($\uparrow$) | LPIPS ($\downarrow$) | Train Step Time (ms) |
|---|---|---|---|---|---|
| TECO (ours) | 48 | **21.9** | **0.703** | **0.157** | **151** |
| MaskGit | 950 | 19.3 | 0.605 | 0.274 | 167 |
| Autoregressive | **44** | 20.1 | 0.640 | 0.197 | 267 |

Table F.3: DMLab dataset comparisons against similar model as TECO without latent dynamics, and Maskgit or AR model on VQ-GAN tokens directly.

Table F.3 shows comparisons between TECO and alternative architectures that do not use latent dynamics. Architecturally, MaskGit and Autoregressive are very similar to TECO, with a few small changes: (1) there is no CNN decoder and (2) MaskGit and AR directly predict the VQ-GAN latents (as opposed to the learned VQ latents in TECO). In terms of training time, MaskGit and AR are a little slower since they operate on $16 \times 16$ latents instead of $8 \times 8$ latents for TECO. In addition, conditioning for the AR model is done using cross attention, as channel-wise concatenation does not work well due to unidirectioal masking. Both models without latent dynamics have worse temporal consistency, as well as overall sample quality. We hypothesize that TECO has better temporal consistency due to weak bottlenecking of latent representation, as a lot of time can be spent modeling likelihood of imperceptible image / video statistics. MaskGit shows very high FVD due to a tendency to collapse in later frames of prediction, which FVD is sensitive to.

## G   METRICS DURING TRAINING

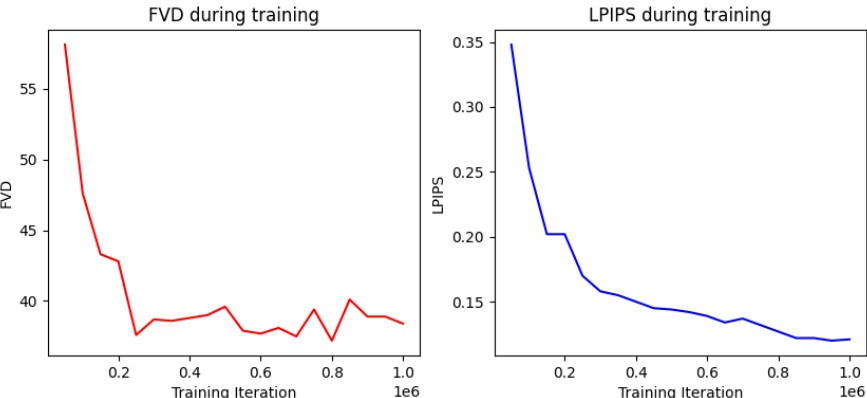

Figure G.1: Comparing FVD and LPIPS evaluation metrics over the course of training. FVD tends to saturate earlier (200k) while LPIPS keeps on improving up until 1M iterations.

Figure G.1 shows plots of FVD (over chunks of generatd 16 frame video) and LPIPS during training, evaluated at saved model checkpoints every 50k iterations over 1M iterations. We can see that although FVD (measuring frame fidelity) tends to saturate early on during training (at around 200k iterations), the long-term consistency metric (LPIPS) continues to improve until the end of training. We hypothesize that this may be due to the model first learning the "easier bits" more local in time, and then learning long-horizon bits once the easier bits have been learned.

## H   PROGRESSION FROM EXISTING WORK

| Model | Architecture | Time per Training Step (s) | FVD↓ | PSNR↑ | SSIM↑ | LPIPS↓ |
|-------|--------------|----------------------------|------|-------|-------|--------|
| VideoGPT / TATS | 3D VQ-VAE + Autoregressive (time + space) | 0.881 | 156 | 11.1 | 0.296 | 0.468 |
| Phenaki | 3D VQ-VAE + MaskGit (time + space) | 0.905 | 725 | 11.0 | 0.202 | 0.474 |
| TECO w/o latent dynamics | 2D VQ-VAE + CNN encoder + Autoregressive (time) + MaskGit (space) | 0.169 | 950 | 19.3 | 0.605 | 0.274 |
| TECO (ours) | 2D VQ-VAE + CNN encoder + Autoregressive (time) + MaskGit (latent) + CNN decoder (space) | **0.131** | **48** | **21.9** | **0.703** | **0.157** |

Table H.1: We iteratively apply architectural modifications starting from existing work up to TECO

Table H.1 shows the progressive improvement from existing work (TATS, Phenaki) and how TECO is able to scale far better on all metrics with our proposed architectural improvements.

# I   HIGH QUALITY SPATIO-TEMPORAL COMPRESSION

| Model | Dataset | FVD↓ |
|-------|---------|------|
| TATS  | DMLab | 54 |
|       | Minecraft | 226 |
| TECO  | DMLab | **7** |
|       | Minecraft | **53** |

Table I.1: Reconstruction FVD comparing TATS Video VQGAN to TECO

Table I.1 compares reconstruction FVD between TECO and TATS. At the same compression rate (same number of discrete codes), TECO learns far better spatio-temporal codes that TATS, with more of a different on more visually complex scenes (Minecraft vs DMLab).

# J   TRADE-OFF BETWEEN FIDELITY AND LEARNING LONG-RANGE DEPENDENCIES

| Downsample Resolution | FVD↓ | PSNR↑ | SSIM↑ | LPIPS↓ |
|-----------------------|------|-------|-------|--------|
| $1 \times 1$ | 44 | **20.4** | **0.666** | **0.170** |
| $2 \times 2$ | 38 | 18.6 | 0.597 | 0.221 |
| $4 \times 4$ | **33** | 17.7 | 0.578 | 0.242 |

Table J.1: Comparing different input resolutions to the temporal transformer

| Latent FDM Arch | FVD↓ | PSNR↑ | SSIM↑ | LPIPS↓ |
|-----------------|------|-------|-------|--------|
| More downsampling + lower resolution computations | 181 | **17.8** | **0.588** | **0.222** |
| Less downsample + higher resolution computations | **94** | 15.6 | 0.501 | 0.277 |

Table J.2: Comparing different Latent FDM architectures with more computation at different resolutions

Table J.1 and Table J.2 show a trade-off between fidelity (frame or image quality) and temporal consistency (long-range dependencies) for video prediction architectures (both TECO, and Latent FDM).

## K FULL EXPERIMENTAL RESULTS

| | TPU-v3 Days | Params | FVD ↓ | PSNR ↑ | SSIM ↑ | LPIPS ↑ |
|---|---|---|---|---|---|---|
| TECO (ours) | 32 | 169M | **27.5 ± 1.77** | **22.4 ± 0.368** | **0.709 ± 0.0119** | **0.155 ± 0.00958** |
| Latent FDM | 32 | 31M | 181 ± 2.20 | 17.8 ± 0.111 | 0.588 ± 0.00453 | 0.222 ± 0.00493 |
| Perceiver-AR | 32 | 30M | 96.3 ± 3.64 | 11.2 ± 0.00381 | 0.304 ± 0.0000456 | 0.487 ± 0.00123 |
| CW-VAE | 32 | 111M | 125 ± 7.95 | 12.6 ± 0.0585 | 0.372 ± 0.000330 | 0.465 ± 0.00156 |
| FitVid | 32 | 165M | 176 ± 4.86 | 12.0 ± 0.0126 | 0.356 ± 0.00171 | 0.491 ± 0.00108 |

Table K.1: DMLab

| | TPU-v3 Days | Params | FVD ↓ | PSNR ↑ | SSIM ↑ | LPIPS ↑ |
|---|---|---|---|---|---|---|
| TECO (ours) | 80 | 274M | 116 ± 5.08 | **15.4 ± 0.0603** | **0.381 ± 0.00192** | **0.340 ± 0.00264** |
| Latent FDM | 80 | 33M | 167 ± 6.26 | 13.4 ± 0.0904 | 0.349 ± 0.00327 | 0.429 ± 0.00284 |
| Perceiver-AR | 80 | 166M | **76.3 ± 1.72** | 13.2 ± 0.0711 | 0.323 ± 0.00336 | 0.441 ± 0.00207 |
| CW-VAE | 80 | 140M | 397 ± 15.5 | 13.4 ± 0.0610 | 0.338 ± 0.00274 | 0.441 ± 0.00367 |
| FitVid | 80 | 176M | 956 ± 15.8 | 13.0 ± 0.00895 | 0.343 ± 0.00380 | 0.519 ± 0.00367 |

Table K.2: Minecraft

| | TPU-v3 Days | Params | FVD ↓ | PSNR ↑ | SSIM ↑ | LPIPS ↑ |
|---|---|---|---|---|---|---|
| TECO (ours) | 275 | 386M | **76.3 ± 1.72** | **12.8 ± 0.0139** | 0.363 ± 0.00122 | 0.604 ± 0.00451 |
| Latent FDM | 275 | 87M | 433 ± 2.67 | 12.5 ± 0.0121 | 0.311 ± 0.000829 | **0.582 ± 0.000492** |
| Perceiver-AR | 275 | 200M | 164 ± 12.6 | **12.8 ± 0.0423** | **0.405 ± 0.00248** | 0.676 ± 0.00282 |

Table K.3: Habitat

| | TPU-v3 Days | Params | FVD ↓ |   | | TPU-v3 Days | Params | FVD ↓ |
|---|---|---|---|---|---|---|---|---|
| TECO (ours) | 640 | 1.09B | 649 ± 16.5 |   | TECO (ours) | 640 | 1.09B | **799 ± 23.4** |
| Latent FDM | 640 | 831M | 960 ± 52.7 |   | Latent FDM | 640 | 831M | 960 ± 52.7 |
| Perceiver-AR | 640 | 1.06B | **607 ± 6.98** |   | Perceiver-AR | 640 | 1.06B | 1022 ± 32.4 |

(a) Using top-k sampling for Perceiver AR and TECO  (b) No top-k sampling

Table K.4: Kinetics

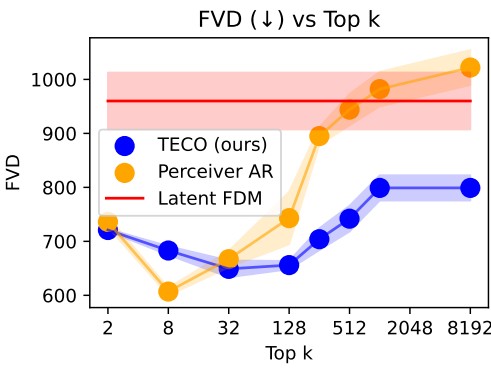

Figure K.1: FVD on Kinetics-600 with different top-k values for Perceiver-AR and TECO

## L    SCALING RESULTS

|  | TPU-v3 Days | Train Seq Len | Params | FVD ↓ | PSNR ↑ | SSIM ↑ | LPIPS ↓ |
|---|---|---|---|---|---|---|---|
| TECO (ours) | 32 | 300 | 169M | **48.2 ± 2.02** | **21.9 ± 0.368** | **0.703 ± 0.0114** | **0.157 ± 0.0119** |
|  |  | 200 | 169M | 59.7 ± 2.29 | 19.9 ± 0.186 | 0.628 ± 0.00821 | 0.187 ± 0.00460 |
|  |  | 100 | 86M | 63.9 ± 7.84 | 15.4 ± 0.199 | 0.476 ± 0.00745 | 0.322 ± 0.00792 |
|  |  | 50 | 195M | 52.7 ± 6.23 | 13.9 ± 0.0311 | 0.418 ± 0.000659 | 0.383 ± 0.000302 |
| Latent FDM | 32 | 300 | 31M | 181 ± 2.20 | 17.8 ± 0.111 | 0.588 ± 0.00453 | 0.222 ± 0.00493 |
|  |  | 200 | 62M | 66.4 ± 3.31 | 17.7 ± 0.114 | 0.561 ± 0.00623 | 0.253 ± 0.00550 |
|  |  | 100 | 80M | 55.6 ± 1.36 | 15.5 ± 0.233 | 0.468 ± 0.00776 | 0.336 ± 0.00511 |
|  |  | 50 | 110M | 68.3 ± 3.19 | 14.0 ± 0.0445 | 0.414 ± 0.424 | 0.385 ± 0.00151 |

Table L.1: DM Lab scaling

|  | TPU-v3 Days | Train Seq Len | Params | FVD ↓ | PSNR ↑ | SSIM ↑ | LPIPS ↓ |
|---|---|---|---|---|---|---|---|
| TECO (ours) | 80 | 300 | 274M | 116 ± 5.08 | 15.4 ± 0.0603 | 0.381 ± 0.00192 | 0.340 ± 0.00264 |
|  |  | 200 | 261M | 109.5 ± 1.46 | 15.4 ± 0.0906 | 0.379 ± 0.00263 | 0.343 ± 0.00148 |
|  |  | 100 | 257M | 85.1 ± 4.09 | **15.7 ± 0.0516** | 0.385 ± 0.00244 | **0.325 ± 0.00121** |
|  |  | 50 | 140M | 80.7 ± 1.42 | 14.8 ± 0.0404 | 0.369 ± 0.00197 | 0.360 ± 0.00133 |
| Latent FDM | 80 | 300 | 33M | 167 ± 6.26 | 13.4 ± 0.0904 | 0.349 ± 0.00327 | 0.429 ± 0.00284 |
|  |  | 200 | 80M | 104.9 ± 3.21 | 15.0 ± 0.0701 | 0.384 ± 0.00320 | 0.366 ± 0.00311 |
|  |  | 100 | 69M | 92.8 ± 4.40 | 15.1 ± 0.0866 | **0.390 ± 0.00281** | 0.358 ± 0.00250 |
|  |  | 50 | 186M | 85.6 ± 2.25 | 14.8 ± 0.0578 | 0.378 ± 0.00144 | 0.372 ± 0.000966 |

Table L.2: Minecraft scaling

## M    RELATED WORK

**Video Generation**    Prior video generation methods can be divided into a few classes of models: variational models, exact likelihood models, and GANs.    SV2P (Babaeizadeh et al., 2017), SVP (Denton & Fergus, 2018), SVG (Villegas et al., 2019), and FitVid Babaeizadeh et al. (2021) are variational video generation methods models videos through stochastic latent dynamics, optimized using the ELBO (Kingma & Welling, 2013) objective extended in time.    SAVP (Lee et al., 2018) adds an adversarial (Goodfellow et al., 2014) loss to encourage more realistic and high-fidelity generation quality.    Diffusion models (Ho et al., 2020; Sohl-Dickstein et al., 2014) have recently emerged as a powerful class of variational generative models which learn to iteratively denoise an initial noise sample to generate high-quality images.    There have been several recent works that extend diffusion models to video, through temporal attention (Ho et al., 2022; Harvey et al., 2022), 3D convolutions (Höppe et al., 2022), or channel stacking (Voleti et al., 2022).    Unlike variational models, autoregressive models (AR) and flows (Kumar et al., 2019) model videos by optimizing exact likelihood. Video Pixel Networks (Kalchbrenner et al., 2017) and Subscale Video Transformers (Weissenborn et al., 2019) autoregressively model each pixel.    For more compute efficient training, some prior methods (Yan et al., 2021; Le Moing et al., 2021; Seo et al., 2022; Rakhimov et al., 2020; Walker et al., 2021) propose to learn an AR model in a spatio-temporally compressed latent space of a discrete autoencoder, which has shown to be orders of magnitudes more efficient compared to pixel-based methods. Instead of a VQ-GAN, Le Moing et al. (2021), learns a frame conditional autoencoder through a flow mechanism.    Lastly, GANs (Goodfellow et al., 2014) offer an alternative method to training video models.    MoCoGAN (Tulyakov et al., 2018) generates videos by disentangling style and motion.    MoCoGAN-HD (Tian et al., 2021) can efficiently extend to larger resolutions by learning to navigate the latent space of a pretrained image generator.    TGANv2 (Saito & Saito, 2018), DVD-GAN (Clark et al., 2019), StyleGAN-V (Skorokhodov et al., 2021), and TrIVD-GAN (Luc et al., 2020) introduce various methods to scale to complex video, such as proposing sparse training, or more efficient discriminator design.

The main focus of this work lies with video prediction, a specific interpretation of conditional video generation.    Most prior methods are trained autoregressive in time, so they can be easily extended to video prediction.    Video Diffusion, although trained unconditionally proposes reconstruction guidance for prediction.    GANs generally require training a separate model for video prediction. However, some methods such as MoCoGAN-HD and DI-GAN can approximate frame conditioning by inverting the generator to compute a corresponding latent for a frame.

**Long-Horizon Video Generation**    CW-VAE (Saxena et al., 2021) learns a hierarchy of stochastic latents to better model long term temporal dynamics, and is able to generate videos with long-term consistency for hundreds of frames.    TATS (Ge et al., 2022) extends VideoGPT which allows for sampling of arbitrarily long videos using a sliding window. In addition, TATs and CogVideo (Hong et al., 2022) propose strided sampling as a simple method to incorporate longer horizon contexts. StyleGAN-V (Skorokhodov et al., 2021) and DI-GAN (Yu et al., 2022) learn continuous-time representations for videos which allow for sampling of arbitrary long videos as well. Brooks et al. (2022) proposes an efficient video GAN architecture that is able to generate high resolution videos of 128 frames on complex video data for dynamic scenes and horseback riding. FDM (Harvey et al., 2022) proposes a diffusion model that is trained to be able to flexibly condition on a wide range of sampled frames to better incorporate context of arbitrarily long videos. Lee et al. (2021) is able to leverage a hierarchical prediction framework using semantic segmentations to generate long videos.

**Long-Horizon Video Understanding**    Outside of generative modeling, prior work such as MeMViT (Wu et al., 2022) and Vis4mer (Mohaiminul Islam & Bertasius, 2022) introduce architectures for modeling long-horizon dependencies in videos.

# N  DATASET DETAILS

## N.1  DMLAB

We generate random $7 \times 7$ mazes split into four quadrants, with each quadrant containing a random combination of wall and floor textures. We generate 40k trajectories of 300 frames, each $64 \times 64$ images. Actions in this environment consist of $20°$ left turn, $20°$ right turn, and walk forward. In order to maximally traverse the maze, we code an agent that traverses to the furthest unvisited point in the maze, with some added noise for stochasticity. Since the maze is a grid, we can easily hard-code a navigation policy to move to any specified point in the maze.

For 3D visualizations, we also collect depth, camera intrinsics and camera extrinsics (pose) for each timestep. Given this information, we can project RGB points into a 3D coordinate space and reconstruct the maze as a 3D pointcloud. Note that since videos are generated only using RGB as input, they do not have groundtruth depth and pose. Therefore, we train depth and pose estimators that are used during evaluation. Specifically, we train a depth estimator to map from RGB frame to depth, and a pose estimator that takes in two adjacent RGB frames and predicts the relative change in orientation. During evaluation, we are given an initial ground truth orientation that we apply sequentially to predicted frames.

Although the GQN Mazes (Eslami et al., 2018) already exists as a video prediction dataset, it is difficult to properly measure temporal consistency. The 3D scenes are relatively simple, and it does not have actions to help reduce stochasticity in using metrics such as PSNR, SSIM, and LPIPS. As a result, FVD is the reliable metric used in GQN Mazes, but tends to be sensitive to noise in video predictions. In addition, we perform 3D visualizations using our dataset that are not possible with GQN Mazes.

## N.2  MINECRAFT

We generate 200k trajectories (each of a different Minecraft world) of 300 $128 \times 128$ frames in the Minecraft marsh biome. We hardcode an agent to randomly traverse the surroundings by taking left, right, and forward actions with different probabilities. In addition, we let the agent constantly jump, which we found to help traverse simple hills, and prevent itself from drowning. We specifically chose the marsh biome, as it contains hilly turns with sparse collections of trees that act as clear landmarks for consistent generation. Forest and jungle biomes tend to be too dense for any meaningfully clear consistency, as all surroundings look nearly identical. On the other hand, plains biomes had the opposite issue where the surroundings were completely flat. Mountain biomes were too hilly and difficult to traverse.

We opt to introduce an alternative to the MineRL Navigate (Guss et al., 2019) since this dataset primarily consists of human demonstrations of people navigating to specific points. This means that trajectories usually follow a relatively straight line, so there are not many long-term dependencies in this dataset, as only a few past frames of context are necessary for prediction.

## N.3  HABITAT

Habitat is a 3D simulator that can render realistic trajectories in scans of 3D scenes. We compile roughly 1400 3D scans from HM3D (Ramakrishnan et al., 2021), MatterPort3D (Chang et al., 2017) and Gibson (Xia et al., 2018), and generate a total of 200k trajectories of 300 $128 \times 128$ frames. We use the in-built path traversal algorithm provided in Habitat to construct action trajectories that allow our agent to move between randomly sampled locations in the 3D scene. Similar to Minecraft and DMLab, the agent action space consists of left turn, right turn, and move forward.

# O    HYPERPARAMETERS

## O.1    VQ-GAN & VAE

|                          | DMLab / Minecraft    | Habitat / Kinetics-600 |
|--------------------------|----------------------|------------------------|
| GPU Days                 | 16                   | 32                     |
| Resolution               | 64 / 128             | 128                    |
| Batch Size               | 64                   | 64                     |
| LR                       | $3 \times 10^{-4}$   | $3 \times 10^{-4}$     |
| Num Res Blocks           | 2                    | 2                      |
| Attention Resolutions    | 16                   | 16                     |
| Channel Mult             | 1,2,2,2              | 1,2,3,4                |
| Base Channels            | 128                  | 128                    |
| Latent Size (VQ-GAN)     | $16 \times 16$       | $16 \times 16$         |
| Embedding Dim (VQ-GAN)   | 256                  | 256                    |
| Codebook Size (VQ-GAN)   | 1024                 | 8192                   |
| Latent Size (VAE)        | $16 \times 16 \times 4$ | $16 \times 16 \times 8$ |

## O.2    TECO

|              | Hyperparameters        | DMLab              | Minecraft          | Habitat            | Kinetics-600       |
|--------------|------------------------|--------------------|--------------------|--------------------|--------------------|
|              | TPU-v3 Days            | 32                 | 80                 | 275                | 640                |
|              | Params                 | 169M               | 274M               | 386M               | 1.09B              |
|              | Resolution             | 64                 | 128                | 128                | 128                |
|              | Batch Size             | 32                 | 32                 | 32                 | 32                 |
|              | Sequence Length        | 300                | 300                | 300                | 100                |
|              | LR                     | $1 \times 10^{-4}$ | $1 \times 10^{-4}$ | $1 \times 10^{-4}$ | $1 \times 10^{-4}$ |
|              | LR Schedule            | cosine             | cosine             | cosine             | cosine             |
|              | Warmup Steps           | 10k                | 10k                | 10k                | 10k                |
|              | Total Training Steps   | 1M                 | 1M                 | 1M                 | 1M                 |
|              | DropLoss Rate          | 0.9                | 0.9                | 0.9                | 0.9                |
| Encoder      | Depths                 | 256, 512           | 256, 512           | 256, 512           | 256, 512           |
|              | Blocks                 | 2                  | 4                  | 4                  | 8                  |
| Codebook     | Size                   | 1024               | 1024               | 1024               | 1024               |
|              | Embedding Dim          | 32                 | 32                 | 32                 | 32                 |
| Decoder      | Depths                 | 256, 512           | 256, 512           | 256, 512           | 256, 512           |
|              | Blocks                 | 4                  | 8                  | 8                  | 10                 |
| Temporal Transformer | Downsample Factor | 8             | 8                  | 4                  | 2                  |
|              | Hidden Dim             | 1024               | 1024               | 1024               | 1536               |
|              | Feedforward Dim        | 4096               | 4096               | 4096               | 6144               |
|              | Heads                  | 16                 | 16                 | 16                 | 24                 |
|              | Layers                 | 8                  | 12                 | 8                  | 24                 |
|              | Dropout                | 0                  | 0                  | 0                  | 0                  |
| MaskGit      | Mask Schedule          | cosine             | cosine             | cosine             | cosine             |
|              | Hidden Dim             | 512                | 768                | 1024               | 1024               |
|              | Feedforward Dim        | 2048               | 3072               | 4096               | 4096               |
|              | Heads                  | 8                  | 12                 | 16                 | 16                 |
|              | Layers                 | 8                  | 6                  | 16                 | 24                 |
|              | Dropout                | 0                  | 0                  | 0                  | 0                  |

| | Hyperparameters | Train Sequence Length (Fewer FLOPs per Frame) | | | |
|---|---|---|---|---|---|
| | | 300 | 200 | 100 | 50 |
| | TPU-v3 Days | 32 | 32 | 32 | 32 |
| | Params | 169M | 169M | 86M | 195M |
| | Resolution | 64 | 64 | 64 | 64 |
| | Batch Size | 32 | 32 | 32 | 32 |
| | LR | $1 \times 10^{-4}$ | $1 \times 10^{-4}$ | $1 \times 10^{-4}$ | $1 \times 10^{-4}$ |
| | LR Schedule | cosine | cosine | cosine | cosine |
| | Warmup Steps | 10k | 10k | 10k | 10k |
| | Total Training Steps | 1M | 1M | 1M | 1M |
| | DropLoss Rate | 0.9 | 0.85 | 0.85 | 0.85 |
| Encoder | Depths | 256, 512 | 256, 512 | 256, 512 | 256, 512 |
| | Blocks | 2 | 2 | 2 | 2 |
| Codebook | Size | 1024 | 1024 | 1024 | 1024 |
| | Embedding Dim | 32 | 32 | 32 | 32 |
| Decoder | Depths | 256, 512 | 256, 512 | 256, 512 | 256, 512 |
| | Blocks | 4 | 4 | 4 | 4 |
| Temporal Transformer | Downsample Factor | 8 | 8 | 2 | 2 |
| | Hidden Dim | 1024 | 1024 | 512 | 1024 |
| | Feedforward Dim | 4096 | 4096 | 2048 | 4096 |
| | Heads | 16 | 16 | 8 | 16 |
| | Layers | 8 | 8 | 8 | 8 |
| | Dropout | 0 | 0 | 0 | 0 |
| MaskGit | Mask Schedule | cosine | cosine | cosine | cosine |
| | Hidden Dim | 512 | 512 | 512 | 768 |
| | Feedforward Dim | 2048 | 2048 | 2048 | 3072 |
| | Heads | 8 | 8 | 8 | 12 |
| | Layers | 8 | 8 | 8 | 8 |
| | Dropout | 0 | 0 | 0 | 0 |

Table O.1: Hyperparameters for scaling TECO on DMLab

| Hyperparameters | | Train Sequence Length (Fewer FLOPs per Frame) | | | |
|---|---|---|---|---|---|
| | | 300 | 200 | 100 | 50 |
| | TPU-v3 Days | 80 | 80 | 80 | 80 |
| | Params | 274M | 261M | 257M | 140M |
| | Resolution | 128 | 128 | 128 | 128 |
| | Batch Size | 32 | 32 | 32 | 32 |
| | LR | $1 \times 10^{-4}$ | $1 \times 10^{-4}$ | $1 \times 10^{-4}$ | $1 \times 10^{-4}$ |
| | LR Schedule | cosine | cosine | cosine | cosine |
| | Warmup Steps | 10k | 10k | 10k | 10k |
| | Total Training Steps | 1M | 1M | 1M | 1M |
| | DropLoss Rate | 0.9 | 0.85 | 0.25 | 0.25 |
| Encoder | Depths | 256, 512 | 256, 512 | 256, 512 | 256, 512 |
| | Blocks | 4 | 4 | 4 | 4 |
| Codebook | Size | 1024 | 1024 | 1024 | 1024 |
| | Embedding Dim | 32 | 32 | 32 | 32 |
| Decoder | Depths | 256, 512 | 256, 512 | 256, 512 | 256, 512 |
| | Blocks | 8 | 8 | 8 | 8 |
| Temporal Transformer | Downsample Factor | 8 | 4 | 2 | 2 |
| | Hidden Dim | 1024 | 1024 | 1024 | 512 |
| | Feedforward Dim | 4096 | 4096 | 4096 | 2048 |
| | Heads | 16 | 16 | 16 | 8 |
| | Layers | 12 | 12 | 12 | 12 |
| | Dropout | 0 | 0 | 0 | 0 |
| MaskGit | Mask Schedule | cosine | cosine | cosine | cosine |
| | Hidden Dim | 768 | 768 | 768 | 768 |
| | Feedforward Dim | 3072 | 3072 | 3072 | 3072 |
| | Heads | 12 | 12 | 12 | 12 |
| | Layers | 6 | 6 | 6 | 8 |
| | Dropout | 0 | 0 | 0 | 0 |

Table O.2: Hyperparameters for scaling TECO on Minecraft

## O.3 LATENT FDM

| Hyperparameters | DMLab | Minecraft | Habitat | Kinetics-600 |
|---|---|---|---|---|
| TPU-v3 Days | 32 | 80 | 275 | 640 |
| Params | 31M | 33M | 87M | 831M |
| Resolution | 64 | 128 | 128 | 128 |
| Batch Size | 32 | 32 | 32 | 32 |
| LR | $1 \times 10^{-4}$ | $1 \times 10^{-4}$ | $1 \times 10^{-4}$ | $1 \times 10^{-4}$ |
| LR Schedule | cosine | cosine | cosine | cosine |
| Optimizer | Adam | Adam | Adam | Adam |
| Warmup Steps | 10k | 10k | 10k | 10k |
| Total Training Steps | 1M | 1M | 1M | 1M |
| Base Channels | 128 | 128 | 128 | 256 |
| Num Res Blocks | 1,1,1,2 | 1,1,2,2 | 1,2,2,4 | 2,2,2,2 |
| Head Dim | 64 | 64 | 64 | 64 |
| Attention Resolutions | 4,2 | 4,2 | 4,2 | 8,4,2 |
| Dropout | 0 | 0 | 0 | 0 |
| Channel Mult | 1,1,1,2 | 1,2,2,2 | 1,2,2,4 | 1,2,3,8 |

Table O.3: Hyperparameters for Latent FDM

| Hyperparameters | Train Sequence Length (Fewer FLOPs per Frame) | | | |
| --- | --- | --- | --- | --- |
| | 300 | 200 | 100 | 50 |
| TPU-v3 Days | 32 | 32 | 32 | 32 |
| Params | 31M | 62M | 80M | 110M |
| Resolution | 64 | 64 | 64 | 64 |
| Batch Size | 32 | 32 | 32 | 32 |
| LR | $1 \times 10^{-4}$ | $1 \times 10^{-4}$ | $1 \times 10^{-4}$ | $1 \times 10^{-4}$ |
| LR Schedule | cosine | cosine | cosine | cosine |
| Optimizer | Adam | Adam | Adam | Adam |
| Warmup Steps | 10k | 10k | 10k | 10k |
| Total Training Steps | 1M | 1M | 1M | 1M |
| Base Channels | 128 | 128 | 128 | 192 |
| Num Res Blocks | 1,1,1,2 | 1,1,2,2,4 | 2,2,2,2 | 3,3,3,3 |
| Head Dim | 64 | 64 | 64 | 64 |
| Attention Resolutions | 4,2 | 4,1 | 4,2 | 8,4,2 |
| Dropout | 0 | 0 | 0 | 0 |
| Channel Mult | 1,1,1,2 | 1,1,2,2,4 | 1,2,3,4 | 1,2,3,4 |

Table O.4: Hyperparameters for scaling Latent FDM on DMLab

| Hyperparameters | Train Sequence Length (Fewer FLOPs per Frame) | | | |
| --- | --- | --- | --- | --- |
| | 300 | 200 | 100 | 50 |
| TPU-v3 Days | 80 | 80 | 80 | 80 |
| Params | 33M | 80M | 69M | 186M |
| Resolution | 128 | 128 | 128 | 128 |
| Batch Size | 32 | 32 | 32 | 32 |
| LR | $1 \times 10^{-4}$ | $1 \times 10^{-4}$ | $1 \times 10^{-4}$ | $1 \times 10^{-4}$ |
| LR Schedule | cosine | cosine | cosine | cosine |
| Optimizer | Adam | Adam | Adam | Adam |
| Warmup Steps | 10k | 10k | 10k | 10k |
| Total Training Steps | 1M | 1M | 1M | 1M |
| Base Channels | 128 | 128 | 128 | 192 |
| Num Res Blocks | 1,1,2,2 | 2,2,2,2 | 3,3,3,3 | 2,2,2,2 |
| Head Dim | 64 | 64 | 64 | 64 |
| Attention Resolutions | 4,2 | 4,2 | 8,4,2 | 8,4,2 |
| Dropout | 0 | 0 | 0 | 0 |
| Channel Mult | 1,2,2,2 | 1,2,3,4 | 1,2,2,3 | 1,2,3,4 |

Table O.5: Hyperparameters for scaling Latent FDM on Minecraft

## O.4  CW-VAE

| Hyperparameters | | DMLab | Minecraft |
|---|---|---|---|
| | TPU-v3 Days | 32 | 80 |
| | Params | 111M | 140M |
| | Resolution | 64 | 128 |
| | Batch Size | 32 | 32 |
| | LR | $1 \times 10^{-4}$ | $1 \times 10^{-4}$ |
| | LR Schedule | cosine | cosine |
| | Optimizer | Adam | Adam |
| | Warmup Steps | 10k | 10k |
| | Total Training Steps | 1M | 1M |
| Encoder | Kernels | 4,4,4 | 4,4,4 |
| | Filters | 256,512,1024 | 256,512,1024 |
| Decoder | Depths | 256,512 | 256,512 |
| | Blocks | 4 | 8 |
| Dynamics | Levels | 3 | 3 |
| | Abs Factor | 6 | 6 |
| | Enc Dense Layers | 3 | 3 |
| | Enc Dense Embed | 1024 | 1024 |
| | Cell Stoch Size | 128 | 256 |
| | Cell Deter Size | 1024 | 1024 |
| | Cell Embed Size | 1024 | 1024 |
| | Cell Min Stddev | 0.001 | 0.001 |

Table O.6: Hyperparameters for CW-VAE

## O.5  FITVID

| Hyperparameters | DMLab | Minecraft |
|---|---|---|
| TPU-v3 Days | 32 | 80 |
| Params | 165M | 176M |
| Resolution | 64 | 128 |
| Batch Size | 32 | 32 |
| LR | $1 \times 10^{-4}$ | $1 \times 10^{-4}$ |
| LR Schedule | cosine | cosine |
| Optimizer | Adam | Adam |
| Warmup Steps | 10k | 10k |
| Total Training Steps | 1M | 1M |
| g Dim | 256 | 256 |
| RNN Size | 512 | 768 |
| z Dim | 64 | 128 |
| Filters | 128,128,256,512 | 128,128,256,512 |

Table O.7: Hyperparameters for FitVid

