# OpenReview forum: "Temporally Consistent Video Transformer for Long-Term Video Prediction"
_ICLR.cc/2023/Conference — Submitted to ICLR 2023_

### Official Review · Reviewer_zBcT · 2022-10-21

**Confidence:** 4
**Correctness:** 3
**Technical Novelty And Significance:** 3
**Empirical Novelty And Significance:** 3
**Recommendation:** 5

**Clarity, Quality, Novelty And Reproducibility:**

The Clarity and Novelty of this paper need to be further justified, which is detailed in the above section. In terms of Reproducibility, this paper provides several resources including code, baseline and dataset.


**Strength And Weaknesses:**

This paper is well-written. The discussion on the previous video prediction works is mostly well done. Their main contribution is extending the short video generation to the long-form video generation while maintaining the temporal consistency. Thus, the motivation is well justified.

They show a quantitative comparison between TECO and other baseline methods in long temporal consistency and sampling speed, demonstrating TECO could achieve competitive results with fast sampling speed.

Besides, this paper introduces three challenging video benchmarks to better measure the long-range consistency in video prediction, which I believe could be a significant contribution to the community.

At last, this paper demonstrates promising performance in these challenging datasets and outperforms the previous state-of-the-art methods.

However, there are also a few points that need to be further explained and addressed.
In Figure 2, it seems that the $\hat x_t = D(z_{t}, h_{t-1})$, which is different from the text in the paper. It seems the prediction of the future time-step from the current time-step include the feature embedding from the future? To make it clear, will be good to show how $\hat x_t $ is computed.  Also, I am not sure how the output of MaskGit ($\hat z_t$) is leveraged in the equation. It will be good to generate notation for $\hat z_t$.

In Encoder, it is not well explained how the proposed algorithm could leverage/benefit from the temporal redundancy in video data. Also the temporal redundancy should be datasets-dependent. There should be some justifications showing there are temporal redundancies with different levels in each dataset.

In addition to these, there are a few minor points:
It will be good to show the formulation of $L_{LPIPS}$ in Eq 3
first row below Eq(6): the decoder is a unsampling -> is an unsampling..
In Figure 3 right, what is the timestep for this figure? As the LPIPS of each method is dynamically changed with timestep (left figure).
In the main paper, some explanation refer the result in the Appendix, which is not suggested.

Suggestion:
In terms of modeling the long-term dependencies, there are a few recent works that are not included in the related work. For example, the MeMViT[1] and ViS4mer[2]. Both of them focus on capturing long-term dependencies in an efficient and effective way. Would be good to at least mention them in the related work.

[1]Memvit: Memory-augmented multiscale vision transformer for efficient long-term video recognition, CVPR 2022


[2] Long movie clip classification with state-space video models, ECCV 2022


**Summary Of The Paper:**

This paper proposes a method, TECO, for the task of video prediction. TECO learns compressed representations to effectively and efficiently condition on long-form videos by modeling long-term dependencies.In order to achieve that, they leverage the MaskGit prior for dynamic prediction and propose a few new architectures based on VQ-GAN. As last, they also introduce several new long-form video prediction benchmarks in this work.


**Summary Of The Review:**


Overall, I would vote weak reject at this point due to some unclear points mentioned in the above section. But I tend to increase my rating after seeing the feedback from authors. As is mentioned in this paper, the long-form video topics have not been well-explored in the community and I believe this paper is able to make contributions in this area.

---

> ### Author Response · Authors · 2022-11-12
> **Response**
>
> Thank you for your review! We address specific comments below.
>
> > Novelty
>
> We refer the reviewer to our global response on novelty. We believe that our work presents several key novel insights in long-horizon video generation through (1) comprehensive benchmarking and (2) technical contributions in architecture design for scaling to long video.
>
> > Clarifications on notation and how samples are generated
>
> Thank you for pointing this out — the figure is correct, and the equations in Eq 5 had an offset in the time subscript that we have corrected. It should be (1) Temporal Transformer $h_t = H(z\leq t)$, (2) Dynamics Prior $p(z_t\mid h_{t-1})$ and (3) Decoder $p(x_t\mid z_t,h_{t-1})$. We have also updated Appendix A with a more detailed description of how sampling is done, and how the MaskGit output is leveraged.
>
> > In Encoder, it is not well explained how the proposed algorithm could leverage/benefit from the temporal redundancy in video data
>
> One large source of redundancy in virtually all video datasets is redundancy in consecutive frames, that is, the previous frame explains the majority of the bits of the next frame. We leverage this through a combination of conditional encoding and decoding, where the model leverages the current latent $z_t$ and temporal history $h_t$ to decode the current frame $x_t$. We have updated the paper to describe this more clearly.
>
> Our ablation in Table F.1 (d) in Appendix F demonstrates that using conditional encodings to compute $z_t$ helps learn more compressed representations to produce better samples.
>
> > it will be good to show the formulation of L_LPIPS
>
> $\mathcal{L}_{LPIPS}$ follows the exact formulation of LPIPS perceptual distance described in the original paper [1]. We have updated the paper to clarify this.
>
> > Clarification on Figure 3
>
> The timestep $t$ for Figure 3 is the $t$-th predicted frame, where the LPIPS measurement for time $t$ is the LPIPS between the groundtruth frame at timestep $t$ and the generated frame at timestep $t$. We show that as the prediction horizon increases, TECO remains the most temporally consistent by retaining the lowest LPIPS metric compared to the ground-truth trajectory, without observing intermediate inputs. The figure was evaluated using generated videos predicting 500 frames into the future.
>
> > Mention MeMVit and ViS4mer
>
> Thank you for the suggestion - we have updated our related works to include these two papers. We would like to note that MeMVit is very similar to the line of work in language models on long sequences — Transformer-XL, Routing Transformer, Compressive Transformer — all of which Perceiver-AR (our baseline) outperforms.
>
> [1] Zhang, Richard, et al. "The unreasonable effectiveness of deep features as a perceptual metric." Proceedings of the IEEE conference on computer vision and pattern recognition. 2018.

---

> ### Comment · Reviewer_zBcT · 2022-12-06
> **Respond to the rebuttal**
>
> Thanks Authors for writing this rebuttal, which has addressed most of my concerns.
> After reading all feedback from other reviewers and the rebuttal, I am happy to change my rating from 5 to 6.

---

### Official Review · Reviewer_7Nbj · 2022-10-25

**Confidence:** 3
**Correctness:** 3
**Technical Novelty And Significance:** 2
**Empirical Novelty And Significance:** 3
**Recommendation:** 6

**Clarity, Quality, Novelty And Reproducibility:**

The paper is generally well-written but it would be great if the authors clarify some parts more. Please read the above review.

**Strength And Weaknesses:**

Strength:
- The paper is generally well-written and easy to follow.
- The ideas presented in the paper are sound. The extensive experiment results support that training with longer sequences by the proposed method is indeed helpful in generating long videos better than the baselines in terms of general fidelity.

Weakness:
1. It is unclear from the paper which factors play a major role in improving the efficiency of video prediction. Based on my understanding, the major improvements are from the spatial downsampling in both the encoder and temporal Transformer, while the model retains the same quadratic complexity as the prior works. Despite the practicality, this is a simple trick that is also applicable to other Transformers, and having it as a major component for enabling long-term prediction limits the significance of the work. The same arguments also apply to DropLoss, given that there are also many hierarchical Transformers that apply the prediction at multiple scales (i.e., I feel that DropLoss is a simplified version of a hierarchical Transformer where the loss is coarsely defined over time).

2. To demonstrate the significance of each component in improving efficiency, it would be great to have ablation studies on complexity (e.g., FLOPS and peak memory) together with fidelity. Also, it would be great to have a comparison with baselines in terms of complexity to demonstrate the full advantages.

3. Since one of the main goals of this work is to improve the long-term consistency in prediction, the evaluation should also clearly demonstrate that part. Current protocols are measuring general fidelity of the predicted frames, and are not specifically focused on long-term consistency. One way to focus on the latter is to condition all video prediction models on ground-truth action trajectories and measure the loss on the part of the trajectories revisiting the observed places. This will isolate how well the model can reconstruct the known places from how good the fidelity of the predicted frames is.

4. When predicting the next frame \hat{x}_t, the decoder takes both the temporal encoding of the Transformer (h_t) and the prior (z_t). It creates a trivial autoencoding path through the prior (x_t → z_t → x_t), which makes it unclear how the model will learn to leverage the temporal encoding h_t.

5. Figure 4 is not quite intuitive to interpret. Based on my understanding, the authors intended to show that TECO is the only method that preserves the size of the environment (which is fixed and bounded), which serves as evidence that it understands the long-term dependency. However, since they all rely on different action sequences, it is hard to isolate its impact. I feel that it would be more intuitive to condition all generative models with the same ground-truth action sequence and show the reconstructed map.


**Summary Of The Paper:**

This paper proposes to improve the discrete sequence model for long-term video prediction. The proposed method operates on (patch-wise) quantized representation of videos extracted by VQ-GAN, and models the temporal dynamics through Transformers. To handle the quadratic complexity of Transformers thus enabling training with long sequences, the authors proposed to incorporate an additional CNN-based encoder and decoder and model the latent dynamics with smaller tokens using temporal and bidirectional Transformers (MaskGit). The proposed method is evaluated on three challenging video benchmarks and demonstrated compelling long-term prediction performance over baselines.

**Summary Of The Review:**

I believe that the paper tackles the interesting and important problem of long-term video prediction, but the technical and empirical significance of each part is unclear from the current draft. It would also be great to have additional experiments to analyze the performance in terms of long-term consistency.

---

> ### Author Response · Authors · 2022-11-12
> **Response [2/2]**
>
> > evaluation should clearly demonstrate long-term consistency in prediction
>
> Our paper proposes to measure temporal consistency through a method similar to what you have described. As described in Section 4.3, we leverage large conditioning context (144 frames, full ground-truth action sequences), and compute PSNR, SSIM, LPIPS of 156 frames of future prediction against the ground-truth continuation. Intuitively, once we have conditioned on enough of the scene and include actions, future predictions should be approximately deterministic. This is done slightly differently than how these metrics are commonly used in existing literature, where usually several futures are sampled and the trajectory with best PSNR / SSIM / LPIPS is selected. However, this does not measure temporal consistency well since we **want** to capture aspects of determinism (consistency in 3D scenes) in our evaluations and not rely on sampling multiple trajectories until we match ground-truth. Our experiments in Appendix D show plots comparing different training sequence lengths (i.e. different context lengths during generation), and we can see a clear trend in PSNR / SSIM / LPIPS as sequence length increases, evidence that this metric does indeed measure long-term consistency.
>
> We considered the exact method you described, but since the action sequence does not fully determine the temporal consistencies in an environment, you are not guaranteed certain correspondences to measure. For example, identical action trajectories in Habitat can generate rooms with different furniture, and thus produce different correspondences. Therefore, we opted for our existing metric which we found to correspond well to long-horizon consistency.
>
>
> > … trivial auto encoding path …
>
> Yes, if $z_t$ has enough capacity to fully encode $x_t$, then $h_t$ will never be used by the decoder. This is not an issue with the architecture, because $h_t$ is still a critical input to the MaskGit that predicts $z_{t+1}$. Regardless, this is rarely the case in practice as we design TECO to leverage temporal redundancies to learn more compressed latents ($x_t$ is $16\times 16$, $z_t$ is $8\times 8$).
>
> Below, we show some ablations demonstrating that $h_t$ is indeed being used (lower FVD, better prediction quality), for separate runs with and without $h_t$ being fed into the decoder. Generally, more information dense scenes (e.g. Minecraft) show more benefit in including $h_t$ than simpler scenes (e.g. DMLab).
>
> | Dataset   | With $h_t$ | Without $h_t$ |
> |-----------|------------|:-------------:|
> | DMLab     | **80**     | 87            |
> | Minecraft | **115**    | 202           |
>
> > Clarification on Figure 4
>
> Thank you for pointing out this concern. Figure 4 serves as an example that TECO (1) generates bounded mazes, and (2) stays more consistent than baselines by preserving exact layout. Our 3D visualizations are best understood through video (found at [https://sites.google.com/view/iclr23-teco/dmlab](https://sites.google.com/view/iclr23-teco/dmlab)) where we see that TECO generates far more reasonable mazes compared to the baselines.
>
> We’ve updated Figure 4 with visualizations on video predictions conditioned on actions and 144 frames. In general, we see that TECO is the only model that consistently retains the maze layout during generation. Latent FDM, Perceiver-AR, and CW-VAE tend to lose context and thus generate larger mazes than in the dataset, and although FitVid stays within the expected bounds, it often produces overlapping walls and self-intersecting grids.

---

> > ### Comment · Reviewer_7Nbj · 2022-11-28
> > **Thanks for the rebuttal**
> >
> > I appreciate the authors for their response. Although I am not convinced that the technical aspect of the work is significant even considering the motivation and practical advantages, I didn't give enough credit to the evaluation part of the paper for assessing long-term consistency. I raised my score to 6.

---

> ### Author Response · Authors · 2022-11-12
> **Response [1/2]**
>
> Thank you for your review! We address specific comments below.
>
> > Novelty
>
> We refer the reviewer to our global response on novelty. We believe that our work presents several key novel insights in long-horizon video generation through (1) comprehensive benchmarking and (2) technical contributions in architecture design for scaling to long video.
>
> > unclear from the paper which factors play a major role in improving efficiency of video prediction…. ablation studies on complexity together with fidelity
>
> Efficiency in TECO comes from two main sources: (1) downsampling VQ codes using an CNN encoder before the temporal transformer, and (2) learning a second level of more compressed VQ codes. As the reviewer has mentioned, downsampling for the temporal transformer helps deal with the quadratic costs of the transformer, or more generally, the prohibitive costs of sequence models over the original input resolution. Importantly, we also find that the compressed codes are important for unlocking more accurate long-term predictions, as they are easier to predict for the MaskGit than low-level VQ codes (see global comment).
>
> Below, we show ablations in DMLab on varying the latent resolution (size of the latent codes), and the downsample resolution (input size to the temporal transformer)
>
> | Low-level VQ Resolution | Temporal Transformer Input Resolution | TPU-v3 Days | FVD | PSNR |  SSIM | LPIPS |
> |:-----------------:|:---------------------:|:-----------:|:---:|:----:|:-----:|:-----:|
> |   $16 \times 16$  |      $1\times 1$      |      55     | 747 | 18.4 | 0.583 | 0.288 |
> |   $16 \times 16$  |      $2\times 2$      |      52     | 323 | 17.5 | 0.536 | 0.287 |
> |    $8 \times 8$   |      $1\times 1$      |      36     |  48 | 21.9 | 0.703 | 0.157 |
> |    $8 \times 8$   |      $2\times 2$      |      40     |  38 | 18.6 | 0.597 | 0.221 |
>
> As can be seen, the low-level VQ resolution and temporal transformer input resolution are both important for efficient modeling of long sequences, where too large latent codes make it easier to fall off the manifold during generation (hence high FVD). In addition, smaller latent codes are more efficient. Smaller downsampling resolutions do bring about some efficiency, but also serve as information bottlenecks to better learn long-horizon dependencies (better PSNR / SSIM / LPIPS for lower downsampling resolutions)
>
> > The same arguments also apply to DropLoss, given that there are also many hierarchical Transformers that apply the prediction at multiple scales (i.e., I feel that DropLoss is a simplified version of a hierarchical Transformer where the loss is coarsely defined over time)
>
> DropLoss and applying losses at multiple timescales are two distinct and orthogonal approaches, where both can be used together, as the former drops out terms (for better learning efficiency) in the loss function, and the latter adds losses at different scales (for better learning coarser dependencies). For example, the original Clockwork-VAE constructs multiple scales of losses over multiple layers of temporal abstraction, and DropLoss can be applied to dropout a random fraction of these  losses at each level of temporal dilation.
>
> > comparison with baselines in terms of complexity
>
> All models trained for each dataset are controlled for compute (TPU-v3 days), with different allocations for each dataset scaling with complexity (DMLab: 32, Minecraft: 80, Habitat: 275, Kinetics: 600). TECO outperforms all baselines across these regimes, showing that it scales more efficiently. Plots in Appendix D also show that TECO scales better than baseline models on different training sequence lengths when controlled for compute.

---

### Official Review · Reviewer_HpHv · 2022-10-25

**Confidence:** 3
**Correctness:** 4
**Technical Novelty And Significance:** 2
**Empirical Novelty And Significance:** 2
**Recommendation:** 5

**Clarity, Quality, Novelty And Reproducibility:**

* Novelty: novelty may be moderate, since TECO is a combination of VQ-GAN, MaskGit, Transformer, CNN encoder and a decoder. The technical contribution may be a whole TECO system that can generate videos (predict future frames conditioned on the current ones) with long-term consistent.
* Quality: Everything come down to Table 1 which report comparisons of TECO with baselines on 4 benchmarks. TECO provides better metric-results. The question is "are these results significant enough to justify publication?" or "do we need user study to confirm the significant improvement brought by TECO"
* Clarity: some minor clarification may be needed. In Fig. 2, the decoder takes z_2 and h_1 as input and predict \hat{x}_2, while in text (at teh end of page 4), it writes \hat{x}_t = D(z_t,h_t), is that be a mismatch of subscripts?


**Details Of Ethics Concerns:**

The is the risk of generating inappropriate content with generative models. But that risk is applied for all generative models, not just only this proposed approach.

**Strength And Weaknesses:**

# Strength
- The proposed TECO has better performance when compared with baselines on standard video prediction metrics.
- Experiments on various benchmarks, baselines are provided.

# Weakness
- It is unclear how significant those improvements (metrics) will render on predicted frames. It would be interesting to conduct a user study to test if those improvements make any difference.


**Summary Of The Paper:**

This paper propose TECO (Temporal Consistent Video Transformer) for long-term video prediction. To make video prediction on long videos feasible, TECO trains on pretrained VQ-GAN codes, using transformer for temporal modeling, and on the top is a decoder (with an expressive prior). Experiments are done on various benchmarks (some are generated from game engine or embodied-AI simulation, e.g., Habitat, one realistic benchmark is also included, e.g., Kinetics-600). Various baselines are also provided. The results show that TECO performs better than other baselines on multiple video prediction metrics. TECO also provides faster sampling while maintain temporal coherence. Written presentation is fair and readable.

**Summary Of The Review:**

Overall, the current submission has moderate novelty and fair experimental results. I currently rate this paper with a borderline rating and would love to hear the authors/other reviewers to discuss about the significance of this work.

---

> ### Author Response · Authors · 2022-11-12
> **Response**
>
> Thank you for your review! We address specific comments below.
>
> > Novelty
>
> We refer the reviewer to our global response on novelty. We believe that our work presents several novel insights in long-horizon video generation through (1) comprehensive benchmarking and (2) technical contributions in architecture design for scaling to long video.
>
> > unclear how significant those improvement will render on predicted frames
>
> Video prediction quality can be viewed on our anonymized website: [https://sites.google.com/view/iclr23-teco](https://sites.google.com/view/iclr23-teco). All samples are randomly sampled, without cherry-picking. Qualitatively, we can see that TECO captures long-horizon dependencies better than baselines. In DMLab, 3D scenes created from TECO video samples produce more coherent mazes than baselines. In Minecraft, although most baselines produce as sharp samples as TECO (Latent FDM, Perceiver-AR), they are less consistent which can be seen in worse correspondence with ground-truth. In Habitat and Kinetics-600, sample quality in baselines tends to deteriorate faster than TECO. Therefore, we believe that our strong improvement in sample quality, along with our discussion on novelty, merits a significant contribution to the existing literature on long horizon video generation.
>
> > Clarify on notation mismatch
>
> Thank you for pointing this out — the figure is correct, and the equations in Eq 5 had an offset in the time subscript that we have corrected. It should be (1) Temporal Transformer $h_t = H(z\leq t)$, (2) Dynamics Prior $p(z_t\mid h_{t-1})$ and (3) Decoder $p(x_t\mid z_t,h_{t-1})$.

---

> > ### Comment · Reviewer_HpHv · 2022-12-06
> > **Thank you for your rebuttal**
> >
> > 1) Thank the authors for providing an anonymous web pages with visualizations. I looked at the videos, and tried a few times but it is very hard for me to find a *significant* difference between TECO and other approaches in both synthetic (DMLab, Minecraft, Habitat) and real (Kinetics-600) videos. I think a proper user study will be much more convincing here.
> > 2) I read the authors' response about the technical novelty. The authors claim (1) proper benchmarking and (2) technical contributions in architecture design for scaling to long video. While I appreciate the authors effort in benchmarking, I am still not convinced about the significance of "downsampling features to enable efficient training over long sequences".
> >
> > Given both technical contribution and experimental evaluation are not convinced enough, I will keep my rating unchanged. I won't argue if other reviewers want to champion this paper. And the authors deserve some credits for benchmarking the problem (I just feel it is not solid enough for a top-tier conference paper).

---

### Official Review · Reviewer_JK8Y · 2022-10-28

**Confidence:** 5
**Correctness:** 4
**Technical Novelty And Significance:** 2
**Empirical Novelty And Significance:** 2
**Recommendation:** 6

**Clarity, Quality, Novelty And Reproducibility:**

Clarity: As discussed in the weakness section above, I think several parts of the method descriptions are not well motivated.

Quality: The video prediction results are impressive. The quantitative evaluation is solid.

Novelty: The method is technically sound, but the novelty is somewhat limited. Compared to the standard temporal transformer for modeling the video frame latent code, the main differences are
- learning a CNN to extract features in consecutive frames
- using MaskGit to decode z_t

Reproducibility: I think the paper includes all sufficient details for reproducibility.

**Details Of Ethics Concerns:**

I don't find specific ethics concerns.

**Strength And Weaknesses:**

Strength:
+ The paper is well-written. The exposition is generally clear.
+ The method is technically sound.
+ The evaluation is thorough and convincing. The paper tested two long-term video prediction tasks (DMLab, Minecraft) and showcases generation results on complex natural video (Kinetics-600). The ablation study in Section 4.5 validates the contributions of various technical components.

Weakness:
- The novelty of the work is somewhat limited. The core difference over prior work on video generation is to first compress the frame latent tokens using an encoder to map concatenated consecutive frame latent code into a new "Vector-Quantized latent dynamics". The main efficiency comes from the strided convolution for downsampling discrete latent z_t and a transposed convolution for upsampling after modeling with a transformer. The resolution downsampling and upsampling operations for transforms have been extensively used in methods like VQVAE and VQGAN.

- In Figure 2 shows the Decoder takes h_1 and z_2 as input. Yet, this contradicts Equation (5) where the decoder models p(x_t | z_t, h_t). Which one is correct?

- Section 3.1 presents the core method. I can follow it up with the temporal transform part since they are common operations. But I did not get the motivations and rationale for designing MaskGit iterative decoding for z_t and a CNN for decoding h_t and z_t to frame latent code x_t. The paper only describes what was done, but does not provide reasons. In particular, I think it would make the exposition clearer if the paper shows a simple baseline method (e.g., from frame latent x_t, run a temporal transformer to predict h_t, and then decode h_t back to x_t) so that the readers understand the core differences and modifications are.

- It would be helpful to show how the sequences of frame tokens are generated in the prediction process. It's hard to imagine what the autoregressive steps are with h_t, z_t, and x_t. I can only guess how it might work but it would very helpful if the paper can show it explicitly. For example, during autoregressive sampling, do we use the decoded \hat(z_t) as input to the decoder?

**Summary Of The Paper:**

This paper presents a method for long-term (up to 300 frames) video prediction. The technical method includes
1) encoding latent code from consecutive frames,
2) a temporal transformer with autoregressive sampling,
3) a MaskGit for decoding the dynamic prior, and
4) a CNN decoder to decode the frame latent code.
To validate the effectiveness of the proposed approach, the paper creates and evaluates several datasets, e.g., DMLab, Minecraft, Habitat, and Kinetics-600. The results show that the proposed method outperforms the state-of-the-art video prediction algorithms.

**Summary Of The Review:**

I think the paper demonstrates great results in long-term video generation. The evaluation is strong. But I am not sure if the technical novelty is sufficient for conferences like ICLR. As discussed earlier, the main source of efficiency for modeling long videos is 1) encoding consecutive frames and 2) downsampling before using a temporal transformer. I thus rate the paper as marginally above the acceptance threshold at this point.

---

> ### Author Response · Authors · 2022-11-12
> **Response**
>
> Thank you for your review! We address specific comments below.
>
> > The novelty of the work is somewhat limited … It would make the exposition clearer if the paper shows a simple baseline method so that readers understand what the core differences and modifications are
>
> We refer the reviewer to our global response on novelty, which also details core modifications from existing work. We believe that our work presents several key novel insights in long-horizon video generation through (1) proper benchmarking and (2) technical contributions in architecture design for scaling to long video.
>
> > Contradiction between Figure 2 and Equation 5
>
> Thank you for pointing this out — the figure is correct, and the equations in Eq 5 had an offset in the time subscript that we have corrected. It should be (1) Temporal Transformer $h_t = H(z\leq t)$, (2) Dynamics Prior $p(z_t\mid h_{t-1})$ and (3) Decoder $p(x_t\mid z_t,h_{t-1})$.
>
> > It would be helpful to show how the sequences of frame tokens are generated in the prediction process
>
> We have updated the paper to include a description of the sampling process in Appendix A. Specifically, videos are generated in a time-autoregressive manner in the vector quantized latent space. Given a sequence of conditioning frames $x_1, \dots, x_t$, we can (1) compute the low-level VQ latents $z_1,\dots, z_t$ (using the pre-trained VQ-GAN encoder, and then the encoder in TECO), (2) feed the latents through the temporal transformer to compute $h_t$, (3) feed $h_t$ to MaskGit to predict the next latent $z_{t+1}$. We then append $z_{t+1}$ to the sequence $z_1,\dots,z_t, z_{t+1}$ and repeat until we get $z_1,\dots,z_t,z_{t+1},\dots,z_T$. If we want to render to RGB, we can decode latents back into frames using the TECO decoder, and then the pretrained VQ-GAN decoder.
>
> > Motivation for MaskGit
>
> We chose MaskGit for the prior since it has shown to be able to model complex distributions, and is fast to sample. Alternatives would be a factorized prior, or an autoregressive prior, both of which we ablate in Appendix F.

---

### Author Response · Authors · 2022-11-12
**Overall Response [1/3]**

We would like to thank all the reviewers for their detailed and insightful reviews. Overall, reviewers highlighted the thorough empirical evaluation but had concerns about novelty. We address this criticism in two ways. First, we highlight the novelty and importance of our comprehensive benchmark, which has not existed for long-term consistent video prediction. Second, we provide new ablations and expand our discussion of insights to clearly isolate the technical novelties of TECO that lead to its substantial empirical gains over previous approaches. We strongly believe that our paper offers an important contribution to the video generation literature and will be of interest to the ICLR community as a whole.

We have updated our submission with edits highlighted in purple, including some clarifications in the main paper, and experiments in the Appendix, all of which are also detailed in our responses below.

# Benchmark Novelty

We emphasize that **there exists very little prior work** on carefully evaluating temporal consistency in long horizon video prediction:

- Most prior work [2,4,5,6] focuses on datasets where short-term dependencies are sufficient for accurate prediction (SkyTimeLapse, TaiChi-HD, UCF-101, FaceForensics, MineRL Navigate). GQN-Mazes [7] has long-horizon dependencies but are difficult to measure due to limited auxiliary data (not enough 3D information, actions, etc.) - FVD tends to capture frame fidelity, whereas PSNR, SSIM , and LPIPS are much less meaningful without action-conditioning. In contrast, we develop new 3 datasets for long-term video prediction based on DMLab, Minecraft, and Habitat. As discussed in Section 4.1, these datasets require long contexts to consistently predict camera movement through complex 3D scenes, such as mazes, complex terrain, and indoor scenes.

- A comparison of the prominent modeling approaches for measuring long-term temporal consistency in video prediction has not existed so far. We rigorously compare the prominent approaches on the these datasets, including an autoregressive transformer (as well as Perceiver-AR for increased training sequence length), MaskGit as used in Phenaki [3], latent diffusion, and latent dynamics models (as well as Clockwork-VAE for increased long-term consistency).

- Prior work often used standard video prediction metrics such as FVD that are more sensitive to visual fidelity than long horizon consistency. In contrast, we provide insights on the usefulness of various evaluation metrics (Appendix C & D) for measuring long-term consistency through computing PSNR / SSIM / LPIPS on large frame and action contexts, and through 3D scene reconstructions derived from the video predictions.

We believe that establishing these datasets together with the performance scores of a wide range of well-tuned architectures posits an important starting point for future work on long-term video prediction.

---

> ### Author Response · Authors · 2022-11-12
> **Overall Response [3/3]**
>
> **Trade-off between fidelity and learning long-range dependencies**
>
> Given a network with fixed capacity, there exists an inherent trade-off between generating high fidelity and temporally consistent videos. We find that long-horizon information can be prioritized through bottlenecking representations, whereas larger representations encourage higher fidelity. Below, we show experiments on DMLab with different low-level VQ resolutions for the temporal transformers (lower resolution means higher bottleneck). To measure fidelity, we compute FVD on short 16-frame sequences, and use PSNR / SSIM / LPIPS on long 300-frame sequences to measure long-horizon dependencies. We see that as the resolution increases (bottleneck decreases), fidelity (FVD) is improved, but a cost of long-horizon consistency (PSNR, SSIM, LPIPS).
>
> | Downsample Resolution | FVD | PSNR |  SSIM | LPIPS |
> |-----------------------|-----|:----:|:-----:|-------|
> | $1 \times 1$          | 44  | **20.4** | **0.666** | **0.170** |
> | $2 \times 2$          | 38  | 18.6 | 0.597 | 0.221 |
> | $4 \times 4$          | **33**  | 17.7 | 0.578 | 0.242 |
>
> To demonstrate that this phenomenon is not limited to TECO, we show that a similar trend exists in our Latent FDM baseline, where models with more computation at higher resolutions tend to have higher fidelity at a cost to temporal consistency.
>
> | Latent FDM Arch                                   | FVD    | PSNR     | SSIM      | LPIPS     |
> |---------------------------------------------------|--------|----------|-----------|-----------|
> | More downsampling + lower resolution computations | 181    | **17.8** | **0.588** | **0.222** |
> | Less downsampling + higher resolution computations  | **94** | 15.6     | 0.501     | 0.277     |
>
> Due to our architecture design and aggressive downsampling, TECO achieves a better trade-off between fidelity and temporal consistency compared to our baseline models, demonstrated by better PSNR / SSIM / LPIPS, in addition to FVD.
>
> **Although frame quality saturates early-on, long-term consistency improves when training longer**
>
> We have added a new ablation in Appendix G, comparing short-horizon evaluation metrics (FVD on 16 frame chunks - measuring frame quality), and long-horizon evaluation metrics (PSNR / SSIM / LPIPS) over the course of training. We observe an interesting phenomenon where short-horizon metrics tend to saturate early on during training at around 200k iterations, but long-horizon metrics (LPIPS) continue to improve up until the end of training at 1M iterations. Therefore, although it may seem like visual sample quality has saturated, the model will continue to improve after longer training. We hypothesize that this may be due to the likelihood objective, where modeling bits from neighboring frames is easier than learning more long-horizon bits scattered throughout the video sequences. This finding motivates the use of an efficient video architecture for TECO, which can be trained for more gradient steps given a fixed computational budget.
>
> **Summary**
>
> In summary, we believe that our submission presents several novel technical and empirical contributions and relevant insights for future work on long horizon video prediction and therefore will be of great value to the community.
>
> **References**
>
> [1] Yan, Wilson, et al. "Videogpt: Video generation using vq-vae and transformers." arXiv preprint arXiv:2104.10157 (2021).
>
> [2] Ge, Songwei, et al. "Long video generation with time-agnostic vqgan and time-sensitive transformer." arXiv preprint arXiv:2204.03638 (2022).
>
> [3] Villegas, Ruben, et al. "Phenaki: Variable length video generation from open domain textual description." arXiv preprint arXiv:2210.02399 (2022).
>
> [4] Skorokhodov, Ivan, Sergey Tulyakov, and Mohamed Elhoseiny. "Stylegan-v: A continuous video generator with the price, image quality and perks of stylegan2." Proceedings of the IEEE/CVF Conference on Computer Vision and Pattern Recognition. 2022.
>
> [5] Saxena, Vaibhav, Jimmy Ba, and Danijar Hafner. "Clockwork variational autoencoders." Advances in Neural Information Processing Systems 34 (2021): 29246-29257.
>
> [6] Harvey, William, et al. "Flexible Diffusion Modeling of Long Videos." arXiv preprint arXiv:2205.11495 (2022).
>
> [7] Eslami, SM Ali, et al. "Neural scene representation and rendering." Science 360.6394 (2018): 1204-1210.

---

> ### Author Response · Authors · 2022-11-12
> **Overall Response [2/3]**
>
> # Technical Novelty
>
> TECO substantially improves long-term consistency in video prediction over SOTA models (video diffusion, FitVid, Perceiver-AR, Clockwork-VAE) by downsampling features to enable efficient training over long sequences. Although the ideas of downsampling for more efficient learning have been explored in other fields, they have not been exhausted in the context of video generation. In fact, **all of our baselines also downsample** in some manner, yet TECO outperforms each of them. In addition to the comprehensive ablations in Appendix D and F, we performed new experiments to provide further insights and understanding.
>
> **Progression from existing work**
>
> We thank Reviewer JK8Y for their suggestion on a further ablation to highlight the core differences from existing work. To clearly isolate the benefits of TECO, we run experiments that interpolate between the closest existing works and TECO on the DMLab dataset. We start from prior video generation work that uses VQ-codes (VideoGPT, TATS), and iteratively apply changes until we get to the TECO architecture. In this experiment, all models use the same training sequence length of 300 frames and were trained for 48 hours on a TPUv3-32.
>
> | Model                    | Architecture                                                                             | Time per Training Step (s) | FVD | PSNR | SSIM  | LPIPS |
> |--------------------------|------------------------------------------------------------------------------------------|----------------------------|-----|------|-------|-------|
> | VideoGPT [1] / TATS [2]  | 3D VQ-VAE + Autoregressive (time & space)                                                | 0.881                      | 156 | 11.1 | 0.296 | 0.468 |
> | Phenaki [3]              | 3D VQ-VAE + MaskGit (time & space)                                                       | 0.905                      | 725 | 11.0 | 0.202 | 0.474 |
> | TECO w/o latent dynamics | 2D VQ-VAE + CNN encoder + Autoregressive (time) + MaskGit (space)                        | 0.169                      | 950 | 19.3 | 0.605 | 0.274 |
> | TECO                     | 2D VQ-VAE + CNN encoder + Autoregressive (time) + MaskGit (latent) + CNN decoder (space) | **0.131**                      | **48**  | **21.9** | **0.703** | **0.157** |
>
> 1. As the comparison shows, approaches like VideoGPT and TATS (autoregressive) and Phenaki (MaskGit) that directly apply a transformer on top of VQ codes are less scalable than the approach of TECO. Applying a transformer directly on the VQ codes requires a lot of memory and compute due to the large number of tokens, preventing models from being scaled up to large capacity without resorting to short training sequences.
>
> 2. Compressing the VQ codes ("TECO w/o latent dynamics") allows more efficient training on longer sequences, resulting in a 5x speedup on the same hardware. As a result, it substantially improves PSNR, SSIM, and LPIPS. However, the architectural bottleneck also makes it more challenging to predict the future VQ codes, resulting in reduced visual fidelity that manifests in a larger FVD score. Empirically, we find FVD to be particularly sensitive to such low-level visual details.
>
> 3. To solve this problem, TECO discretizes the CNN encoder output to obtain more compressed high-level VQ codes that the transformer predicts, which are then mapped back to low-level VQ codes and finally images. Predicting the more compact high-level VQ codes allows TECO to efficiently train on long sequences while retaining sharp generations, and in this way substantially outperform previous approaches across the four metrics.
>
> **High quality spatio-temporal compression**
>
> While prior works such as VideoGPT [1] and TATS [2] (an improved version of VideoGPT) spatiotemporally compress data using a 3D VQ-VAE, they leave a lot of room for improved compression and efficiency. Below, we compute the FVD of reconstructions comparing TATS to TECO at the same compression level (8 x 8 x 300 = 19,200 low-level tokens for a 300 frame video). **At the same compression rate**, TECO reconstructs frames with higher fidelity (lower FVD) than prior methods, and the improvement is exaggerated on the more visually complex dataset (Minecraft):
>
> | Model                   | Dataset |    FVD  |
> |-------------------------|-----------|:------:|
> | TATS [2] | DMLab   | 54
> |                         | Minecraft    | 226    |
> | TECO                    | DMLab        | **7**  |
> |                         | Minecraft          | **53** |

---

### Author Response · Authors · 2022-11-18
**Further Rebuttal Discussion**

We thank the reviewers for taking the time to review our paper. We believe that our rebuttal has addressed the concerns brought forth by the reviewers, and would greatly appreciate it if reviewers could adjust their scores if their concerns have been met, or further discuss any remaining concerns.

---

### Decision · Program_Chairs · 2023-01-20

**Decision:**

Reject

**Justification For Why Not Higher Score:**

The concern on the limited technical novelty is critical, and was not successfully addressed during the rebuttal/discussion period.

**Justification For Why Not Lower Score:**

N/A

**Metareview: Summary, Strengths And Weaknesses:**

The reviewers generally appreciated that the paper is well-written with clear motivations. They also appreciated the extensive evaluation and the proposed benchmarks.

However, there was an unanimous concern on the technical novelty. They pointed out that the proposed approach is a combination of existing, well-known ideas and the core component (down/upsampling) has been used extensively in prior work. It was also mentioned that there is no clear ablation results showing contribution of different components of the approach. The authors rebutted by saying that, although the idea of downsampling has been explored before, it has not been demonstrated extensively in long-term video prediction task. They also provided some additional experimental results further showing the contribution of different components of the approach. Unfortunately, even after the rebuttal the reviewers were still unconvinced by the significance of the claimed technical novelty.

This meta reviewer believes that the paper provides strong empirical results, but share the same concern with the reviewers that the technical novelty isn't strong enough to warrant acceptance. Therefore, we recommend rejection at this time.